# The Intrinsic Dimension of Prompts in Internal Representations of Large Language Models

## Abstract

We study the geometry of token representations at the prompt level in large language models through the lens of intrinsic dimension. Viewing transformers as mean-field particle systems, we estimate the intrinsic dimension of the empirical measure at each layer and demonstrate that it correlates with next-token uncertainty. Across models and intrinsic dimension estimators, we find that intrinsic dimension peaks in early to middle layers and increases under syntactic and semantic disruption (by shuffling tokens), and that it is strongly correlated with average surprisal, with a simple analysis linking logits geometry to entropy via softmax. As a case study in practical interpretability and safety, we train a linear probe on the per-layer intrinsic dimension profile to distinguish malicious from benign prompts before generation. This probe achieves accuracy of 90 to 95% in different datasets, outperforming widely used guardrails such as Llama Guard and Shield Gemma. We further compare against linear probes built from layerwise entropy derived via the Tuned Lens and find that the intrinsic dimension-based probe is competitive and complementary, offering a compact, interpretable signal distributed across layers. Our findings suggest that prompt-level geometry provides actionable signals for monitoring and controlling LLM behavior, and offers a bridge between mechanistic insights and practical safety tools.

## 1 Introduction

Large language models (LLMs) make predictions by iteratively refining token representations across layers. Understanding how these internal representations evolve and what they reveal about the underlying model remains a core challenge for interpretability and safety. We take a geometric perspective, grounded in recent analytic views of transformers as mean-field interacting systems, where token dynamics is governed by their empirical measure, i.e., the layerwise distribution of token representations viewed as a point cloud in embedding space (Vuckovic et al., 2020; Geshkovski et al., 2024a;b;c; Sander et al., 2022). In particular, Geshkovski et al. (2024b) showed that phenomena like rank collapse can be understood by studying the geometry of tokens, motivating further analysis. We propose prompt-level intrinsic dimension (ID), a measure of the effective dimensionality of the token point cloud, to probe the empirical measure at each layer. We aim to show that it directly characterizes the internal geometry that governs token interactions and that it encodes information directly usable for downstream tasks.

Our approach is inspired by geometric studies that have measured dataset-level properties using last-token representations (Valeriani et al., 2023; Cheng et al., 2023; Skean et al., 2024; Acevedo et al., 2024) to locate semantic phases across layers. However, last-token analyses obscure intra-prompt structure and do not directly probe the empirical measure governing token dynamics. We instead compute ID at the prompt level, i.e. using all tokens within a prompt. This allows us to connect geometry to model uncertainty at the resolution where interactions actually occur.

First, we provide qualitative evidence that prompt-level geometry tracks semantic organization. Across models, the intrinsic dimension exhibits a characteristic peak in early-to-middle layers, a behaviour that also appears in dataset level representation geometry (Cheng et al., 2024). When we disrupt syntax and semantics by shuffling tokens, this peak increases, indicating higher-dimensional geometry. We then show that intrinsic dimension is quantitatively related to model uncertainty: the prompt-level ID correlates with the model's average surprisal (i.e., next-token cross-entropy), and

the correlation appears already in early layers. We explain this through an analysis consisting of three steps: last-layer hidden states map linearly to logits, and softmax entropy increases with the effective dimensionality of the logit manifold. We demonstrate this fact analytically in toy settings and empirically via contextual entropy.

To demonstrate that these insights are directly useful for downstream tasks, we present an experiment in pre-output safety screening. We extract a per-layer ID feature vector for each prompt and train a linear classifier to distinguish benign from malicious prompts. This simple geometric signal achieves 90–95% accuracy, outperforming Llama Guard (Inan et al., 2023) and Gemma Shield (Zeng et al., 2024) on the same data. For comparison, we also build a different data vector using tuned-lens to unembed next token predictions and computing entropy on them. This technique shows comparable performance to the ID feature vector, confirming that internal geometry and latent uncertainty provide complementary, actionable safety signals before generation.

Overall, we provide the following contributions:

- We introduce prompt-level intrinsic dimension as a probe of the empirical measure that governs token dynamics, and map its evolution across layers.
- We establish a layerwise correlation between geometry (ID) and uncertainty (average surprisal), with supporting theoretical intuition from logits–softmax geometry.
- We provide a practical case study of pre-output safety screening, where a linear classifier on ID features separates malicious from benign prompts with 90–95% accuracy, outperforming standard safety tools, and corroborated by tuned-lens entropy
- We clarify how prompt-level geometry differs from dataset-level last-token analyses, and release code to reproduce all results.

Together, these findings position token-geometry as a unifying lens: it summarizes how transformers organize contextual information, predicts uncertainty without accessing outputs, and enables simple, effective, pre-output interventions for interpretability and safety.

## 2    RELATED WORK

**Analytic Approaches to Transformer Models.** Recent analytical works (Geshkovski et al., 2024b; 2023; Castin et al., 2024; Cowsik et al., 2024) indicate that analyzing geometric properties of token representations at the prompt level and their dynamics can offer meaningful insights into how transformers function by viewing the evolution of tokens in the transformer layers as particles in a dynamical system. This perspective not only offers insights into the geometric dynamics of tokens but also addresses the trainability of transformers based on initialization hyperparameters, including the strength of attentional and MLP residual connections. This analytical framework highlights the significance of studying the distribution of the internal representations of the tokens (referred to as the *empirical measure*) by i) suggesting a relation between the empirical measure to the next token prediction (Geshkovski et al., 2024b) ii) understanding the role of the empirical measure in governing the token dynamics (Agrachev & Letrouit, 2024).

**Geometric Approaches to Transformer Models.** The manifold hypothesis posits that real-world high-dimensional data often lie on or near a lower-dimensional manifold within the high-dimensional space (Goodfellow et al., 2016). The dimension of this approximating manifold is usually named the *intrinsic dimension* of the data. Several studies have demonstrated that the intrinsic dimension of data representations in deep networks shows a remarkable dynamic range, characterized by distinct phases of expansion and contraction (Ansuini et al., 2019; Doimo et al., 2020; Pope et al., 2021). Data manifolds created by internal representations in deep networks have been also explored from the perspective of neuroscience and statistical mechanics (Chung et al., 2018; Cohen et al., 2020). In LLMs, a geometric analysis of representations has uncovered a rich set of phenomena. Geometric properties, such as intrinsic dimension and the composition of nearest neighbors, evolve throughout the network's sequence of internal layers. These changes mark distinct phases in the model's operation, signaling the localization of semantic information (Valeriani et al., 2023; Cheng et al., 2023; Skean et al., 2024). Acevedo et al. (2024) analyze the intrinsic dimension by considering all tokens to reveal semantic correlations in images and text inside deep neural networks. While the aforementioned works analyze internal representations in linguistic processing, the geometry of context embeddings

has been linked to language statistics (Zhao et al., 2024) and used to highlight differences between real and artificial data (Tulchinskii et al., 2023). Complementary views include information-geometry analyses that track higher-order structure via cumulant expansions (Viswanathan & Park, 2025), and topological data analysis that follows cross-layer evolution to characterize persistent features (Gardinazzi et al., 2024).

**Safety in LLMs.** Existing literature dealing with LLM safety typically targets two related threats: jailbreaks and prompt injections. In both cases, an attacker's text can override guardrails or combine untrusted with trusted instructions, forcing a model to produce harmful content, or driving it to take unintended actions (for example, leaking data or misusing tools (Greshake et al., 2023; Zou et al., 2023)). Public benchmarks are useful for comparing defenses, but many have well-known issues: i) they quickly become obsolete as models and attacks evolve, ii) labels may mix up toxicity/politics with true attacks and iii) engineered prompts often dominate, which do not resemble well real-life usage (Chao et al., 2024; Mazeika et al., 2024; Schulhoff et al., 2023) (see also this recent article (Stangl & Davis, 2025)). Usable systems commonly rely on moderation and safety classifiers, such as Llama Guard (Inan et al., 2023) and ShieldGemma (Zeng et al., 2024)) and on probability-based detectors such as GLTR (Gehrmann et al., 2019) or DetectGPT (Mitchell et al., 2023), which look for statistical anomalies in text. These baselines are practical but can miss paraphrased or domain-specific attacks and may degrade under distribution shift (see Huang et al. (2024) for a comprehensive review).

## 3 METHOD

Transformer models take as input a sequence of vectors embedded in $d$-dimensions of varying length $N$, $\{x_i\}_{i \in [N]} \in \mathbb{R}^{d \times N}$. Each element of the sequence is called a *token*, while the entire sequence is a *prompt*. A transformer is then a sequence of maps:

$$\{x_i(1)\}_{i \in [N]} \rightarrow \{x_i(2)\}_{i \in [N]} \cdots \rightarrow \{x_i(N_{\text{layers}})\}_{i \in [N]}, \tag{1}$$

where $x_i(\ell) \in \mathbb{R}^{d \times N}$ represents the $i$-th token at layer $\ell$, $N_{\text{layers}}$ the total number of model layers and $N$ is the number of tokens.

In transformer models, prompts can vary based on the specific application, representing protein sequences, image pixels, or text sentences. In this study, we focus on causal language models and use sentences as our input prompts, though the technique can be extended to other input types as well. The prompt size can significantly vary depending on the dataset considered: sentences can be $\mathcal{O}(10)$ - $\mathcal{O}(1000)$ tokens long. Given that our goal is to study and interpret the geometrical behavior at the token level across model layers, we select prompts with a sufficient number of tokens, i.e. $N \gtrsim 500$ tokens, to ensure reliable estimates of our observables.

**Empirical measure.** Given $n$ points at positions $x_1, \ldots, x_n \in \mathbb{R}^d$ (a point cloud), their empirical measure is the probability measure $\mu = \frac{1}{n} \sum_{j=1}^{n} \delta_{x_j}$, i.e., the empirical measure encodes the distribution of points in the embedding space. In the context of transformers (Geshkovski et al., 2024b), the empirical measure characterizes the distribution of the tokens at each layer of the sequence 1. The empirical measure for the last layer is the *output* measure. The dynamical evolution of tokens in this framework, as described by Equation (1) in Agrachev & Letrouit (2024), indicates that the change in the token representation of token $i$ is controlled by a layer-dependent kernel $K_\ell$ and depends purely on the current token representation $x_i(\ell)$ and the empirical measure[1]. To probe the empirical measure across layers, we use the intrinsic dimension, as defined below.

### 3.1 INTRINSIC DIMENSION ESTIMATORS

We estimate the intrinsic dimension (ID) of each prompt's token cloud using kNN-based estimators as they are comparatively robust to high ambient dimensionality and can capture nonlinear manifold structure by operating locally rather than fitting a global linear subspace. These methods typically assume locally approximately uniform density (often modeled by a Poisson process) and small curvature at the neighborhood scale, and are naturally multiscale via the choice of $k$. Concretely,

---

[1]The dynamics of a token $i$ depends on the position of all the tokens $x_j(\ell)$ but not on their labels, which is an assumption in the mean-field interacting particle framework.

we use: GRIDE (Denti et al., 2022), a likelihood-based estimator from kNN distance ratios; ESS (expected simplex skewness) (Johnsson et al., 2015), which maps the shape of local simplices to ID via closed-form expectations under isotropy; and TLE (tight local ID estimation) (Amsaleg et al., 2019), which fits the tail behavior of kNN distances with bias-reduced corrections for small samples. For ESS and TLE, we report layerwise prompt-level IDs by averaging local tokenwise estimates, whereas GRIDE directly produces a single prompt-level ID via maximum-likelihood using the ratio of distances to the nearest neighbours.

**GRIDE.** GRIDE (Denti et al., 2021) is a likelihood-based ID estimator that estimates the intrinsic dimension $\hat{d}(n_1, n_2)$ using the ratios $\dot{\mu} = \mu_{i,n_1,n_2} = \frac{r_{i,n_2}}{r_{i,n_1}}$, where $r_{i,k}$ is the Euclidean distance between point $i$ and its $k$-th nearest neighbour and $1 \leq n_1 < n_2$. Under the assumption of local uniform density, the distribution of $\mu_{i,n_1,n_2}$ is given by,

$$f_{\mu_{i,n_1,n_2}}(\dot{\mu}, d) = \frac{d\left(\dot{\mu}^d - 1\right)^{n_2 - n_1 - 1}}{\dot{\mu}^{(n_2-1)d+1} B\left(n_2 - n_1, n_1\right)}, \quad \dot{\mu} > 1 \tag{2}$$

where $B(\cdot, \cdot)$ is the beta function. The ID estimate $\hat{d}(n_1, n_2)$ is obtained by maximizing the above likelihood with respect to $d$ assuming that the ratios $\mu_{i,n_1,n_2}$ are independent for different points. The conventional choice for the GRIDE algorithm is to set $n_2 = 2n_1$ and examine the variation of $\hat{d}$ for $n_2 \in \{2, 4, 8..\}$, where the parameter $n_2$ is known as the range scaling parameter. The prompts we analyze have $N = 1024$ tokens and in Fig. 6 we check the dependence of ID estimate on range scaling $\in \{2, 4, 8, ..256\}$ for prompt 3218, which suggests the choice of scaling = 4. We additionally check for the Pareto distribution of log ratios in Appendix A (Facco et al., 2017; Denti et al., 2022), that confirms the validity of the intrinsic dimension estimates from GRIDE.

**TLE.** The Tight Local intrinsic dimensionality Estimator (Amsaleg et al., 2018; 2019) algorithm is a nearest neighbor based local ID estimator that is obtained using maximum likelihood estimate techniques over all available pairwise distances among the members of the neighborhood.

**ESS.** The Expected Simplex Skewness (ESS) (Johnsson et al., 2015) is a nearest neighbor based local ID estimator that constructs a simplex with one vertex at the centroid of the local dataset and other vertices at the neighbors. The expected simplex skewness is defined pointwise as the ratio of the volume of the simplex over the volume if all the edges to the centroid were orthogonal.

For the ESS and TLE estimators, we use $k = 10$ and $k = 20$ following Cheng et al. (2023) and use the implementation provided in Bac et al. (2021). In the main text, we quote results for GRIDE only, since all three methods give qualitatively similar results. In Appendix A, we assess the local homogeneity assumption required by our kNN-based ID estimators using Point Adaptive kNN method (Rodriguez et al., 2018), showing that token neighborhoods remain approximately constant density up to $k^* \sim 20$ across layers.

### 3.2 MODELS AND DATASETS

**Models.** In sections 4 and 5, we analyze 4 different pre-trained decoder-only LLMs: Llama 3 8B (Meta, 2024), Mistral 7B (Jiang et al., 2023), Pythia 6.9B (Deduplicated) (Biderman et al., 2023) and Opt 6.7B (Zhang et al., 2022) each of them having 32 hidden layers and a hidden dimension of 4096. We use the PYTHIA models that were trained on the Pile after the dataset has been globally deduplicated. For brevity, we call them LLAMA, MISTRAL, PYTHIA and OPT from now on. In the plots, layer 0 represents the embedding layer, with the hidden layers starting from layer 1. We extract internal representations from these models using the HuggingFace Transformers library[2]. The token representations, stored in the `hidden_state` variable, correspond to the representations in the residual stream (Elhage et al., 2021) after one attention and one MLP update. In the models considered, layer normalization is applied before self-attention and MLP sublayers. LLAMA, MISTRAL and OPT add the self-attention outputs to the residual stream before the MLP whereas PYTHIA adds the self-attention and MLP sublayer outputs to the residual stream in parallel. In Appendix F, we use TunedLens[3] (Belrose et al., 2023) to obtain the entropy of the latent predictions for the GPT-2

---

[2]Link to the library: https://huggingface.co/docs/transformers

[3]Link to the repository: https://github.com/AlignmentResearch/tuned-lens

(Radford et al., 2019) models (small, large and XL), PYTHIA models (160M, 410M, 2.8B and 6.9B), OPT and LLAMA.

**Datasets.** As a dataset representative of text in an extensive way, we use the Pile dataset, which comprises text from 22 different sources (Gao et al., 2020). For computational reasons, we opted for the reduced size version Pile-10K (Nanda, 2022). We further filter only prompts of sequence length $N \geq 1024$ according to the tokenization schemes of all the above models. This choice ensures a reliable estimate of ID. This results in 2244 prompts after filtering. We truncate the prompts by retaining the first $N = 1024$ tokens to eliminate the length-induced bias in our ID estimates, if it were to be present. We describe datasets that are specific to the downstream task experiment of pre-output screening in Section 6.

## 4 QUALITATIVE DESCRIPTION OF INTRINSIC DIMENSION OF PROMPTS

We examine the intrinsic dimension profile of tokens as a function of layers. For a qualitative understanding of the intrinsic dimension at the prompt level, we plot the intrinsic dimension for a given prompt and its shuffled version. By disrupting the syntactic and semantic structure while preserving unigram frequency distribution, we observe the effect of shuffling on the intrinsic dimension profile across layers for different models.

**Shuffling method.** We define the shuffling of tokens in the following way: given a prompt with $N$ tokens, $X = \{x_i\}_{i \in [N]}$, we split the sequence into $nBlocks$ blocks of size $B$ such that $nBlocks \times B = N$ and take one random permutation of the blocks, as schematically presented in Figure 7. Note that the shuffle index for the fully shuffled case ($\hat{S}$) corresponds to the value of $S$ when the number of tokens $N = 4^{\hat{S}}$. In Appendix B we include the shuffling algorithm and a schematic example of the shuffling method.

We show two main results: i) the effect of various degrees of shuffling on our metrics for a single, random prompt and ii) the qualitative behavior of the unshuffled and the fully shuffled prompts on average. For the former observable, we consider the the $3218^{\text{th}}$ prompt from the Pile-10K dataset, with the Pile set name: *ArXiv*. This prompt is shuffled to six different levels labeled by ($S = 0, 1, \ldots, 5$) where the shuffle index $S$ quantifies the degree of shuffling: $S = 0$ represents the unshuffled state, while $S = 5$ corresponds to the fully shuffled case. We study the representations of this prompt using representations from LLAMA. For the average behavior, we find the averages of the ID over 2244 prompts. Figure 1 displays the ID calculated for a range scaling of 4 for LLAMA. The Left Panel

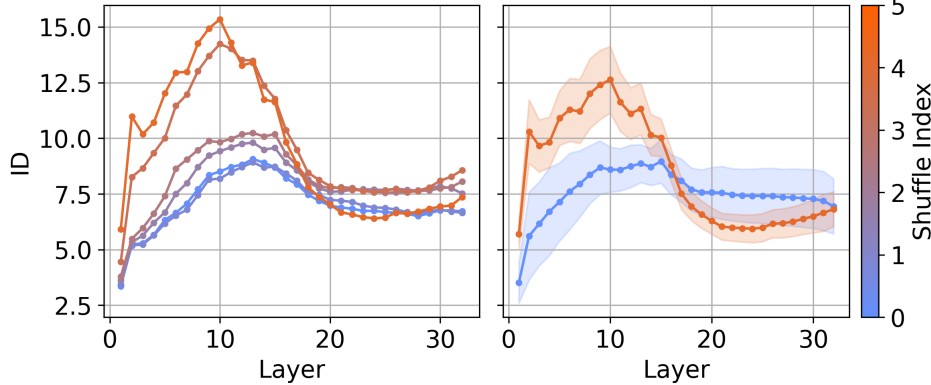

Figure 1: **Intrinsic Dimension.** Left Panel: intrinsic dimension for a single random prompt as a function of model layers. Right Panel: intrinsic dimension averaged over 2244 prompts as a function of layers for the full shuffle ($S = 5$) and the structured case ($S = 0$). The shaded regions indicate the standard deviation from the mean. The color bar indicates the shuffle index $S$. All curves have been calculated for the LLAMA model.

shows the ID profile of a single prompt at various levels of shuffling, while the Right Panel presents the average ID across 2244 prompts for both fully shuffled and structured cases. In all scenarios, we

observe a peak in ID in the early to middle layers. Additionally, the height of this peak increases with the degree of shuffling, indicating a correlation between the two. We show results for other models and estimators in Appendix C.

**Relation to previous work.**  Acevedo et al. (2024) examine how the semantic content of representations influences their intrinsic dimension, suggesting that representations with shared semantic content exhibit a lower intrinsic dimension. This explains the lower intrinsic dimension for the unshuffled prompt. By contrast, prior studies at the dataset level using last-token representations report the opposite shuffling effect: Cheng et al. (2024) find that shuffled prompts yield lower ID than unshuffled ones. This discrepancy arises because dataset-level analyses aggregate last tokens across many prompts, where semantic alignment is weak, inflating ID (typically $\mathcal{O}(40)$) relative to prompt-level token clouds ($\mathcal{O}(10)$). In other words, dataset-level ID primarily reflects the global syntactic/semantic mix of the corpus (Cheng et al., 2024), whereas prompt-level ID captures intra-prompt semantic correlations (Acevedo et al., 2024). We analyze these differences in more detail in Appendix D.

## 5 INTRINSIC DIMENSION IS CORRELATED WITH THE MODEL'S LOSS

In this section, we examine the correlation between the layerwise intrinsic dimension of prompts and their average surprisal of the next token prediction. Given a prompt $X = (x_1, \ldots, x_N)$ and the model's next token prediction $p_\theta$ over a vocabulary $\mathcal{V}$, the average surprisal[4] is

$$\text{average surprisal}(X) = -\frac{1}{N} \sum_{i}^{N} \log p_\theta \left( x_i \mid x_{<i} \right) \tag{3}$$

where $\log p_\theta \left( x_i \mid x_{<i} \right)$ is the log-likelihood of the $i^{\text{th}}$ token conditioned on the preceding tokens ($x_{<i}$). For the population of 2244 prompts across different models used in the previous section, we evaluate the correlation between the $\log$ ID and average surprisal$(X)$ for each layer using the Pearson correlation coefficient ($\rho$), defined as the ratio between the covariance of two variables and the product of their standard deviations.

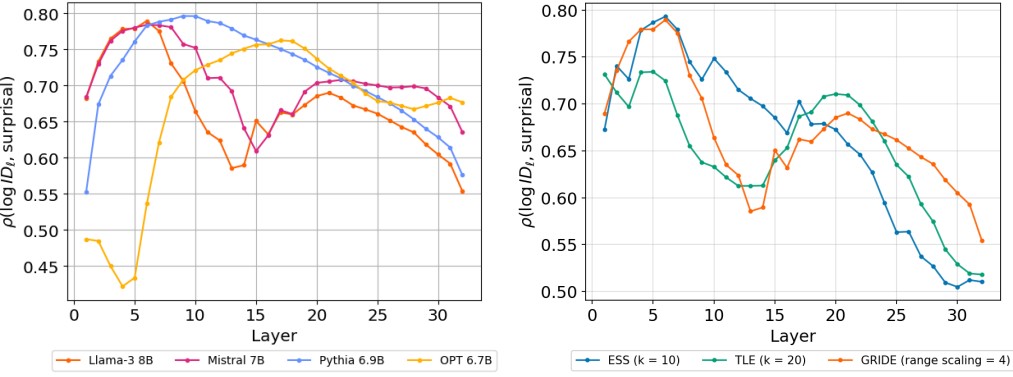

Figure 2: **Correlation between intrinsic dimension and the average surprisal.** Pearson coefficient between the logarithm of the intrinsic dimension and model surprisal for different models and estimators as a function of layers. Left Panel: The four curves correspond to LLAMA (orange), MISTRAL (magenta), PYTHIA (blue), and OPT (yellow). The intrinsic dimension was calculated for the GRIDE estimator at scaling = 4, refer to Figure 12 for scaling = 2 and 8. Right Panel: The three curves correspond to three intrinsic-dimension estimators: ESS ($k$=10), TLE ($k$=20), and GRIDE (range scaling = 4), calculated for LLAMA. We observe a positive correlation between intrinsic dimension and surprisal across all models and estimators starting from the early layers. The $p$-values for the Pearson coefficients in this plot are all below $0.01$.

---

[4]This quantity is referred to by various names in the literature, including average cross-entropy loss, log perplexity, and average next-token prediction error, among others.

As shown in left panel of Figure 2, all four models have a high positive correlation across the layers of the model, implying that prompts with a higher average surprisal have a higher intrinsic dimension. This behaviour is replicated across different estimators as well (right panel), and across choices of range scalings for GRIDE, as shown in Figure 12 in Appendix E. This observation is qualitatively consistent with the shuffling experiment in Section 4, the shuffled data is expected to have a higher surprisal and hence a higher intrinsic dimension. Notably, this correlation exists in the early layers even though the surprisal is calculated after the final layer. This can be attributed to the substantial Pearson correlation coefficient ($\rho > 0.6$) between the intrinsic dimension of the internal layers and the last layer. We empirically check this in Figure 13 in Appendix E where we also find a strong correlation between the intrinsic dimension of adjacent layers.)

**Why does intrinsic dimension track uncertainty?** The observed correlation between a geometric quantity (the ID of internal representations) and an information-theoretic quantity (the surprisal) occurs at the softmax layer between the last layer representations and the next token predictions. It is therefore worth discussing the relationship between the ID at the last layer and the surprisal in more detail. We can summarize the idea with the following steps:

1. **Unembedding Tokens to Logits**: We expect the ID of the last layer to be strongly correlated to the ID of the logits since the unembedding is a linear transformation. This is confirmed by the Pearson coefficient of $\rho = 0.98$ between the log ID of the last layer and the logits.

2. **Logits to Contextual Entropy**: Here we relate the geometric perspective to the information-theoretic perspective. In this context, a softmax layer converts the logits to next token prediction probabilities, $p_\theta (v \mid x_{<i})$. From this, the contextual entropy (Wilcox et al., 2023) is defined as

$$H (x_{<i}) = - \sum_{v \in \mathcal{V}} p_\theta (v \mid x_{<i}) \log p_\theta (v \mid x_{<i}) = \underset{v \sim p(\cdot \mid \boldsymbol{x}_{<i})}{\mathbb{E}} \left[ - \log p_\theta (v \mid x_{<i}) \right] \quad (4)$$

where $(x_{<i})$ is the *context*. We can average this quantity over all the tokens in a prompt to obtain the *average contextual entropy* $\mathcal{H}(X)$:

$$\mathcal{H}(X) = \frac{1}{N} \sum_{i=1}^{N} H (x_{<i}) = - \frac{1}{N} \sum_{i=1}^{N} \sum_{v \in \mathcal{V}} p_\theta (v \mid x_{<i}) \log p_\theta (v \mid x_{<i}) \quad (5)$$

We empirically show a correlation between the logarithm of logits ID and the contextual entropy in the LLAMA model by observing a Pearson correlation of $\rho = 0.6$, as shown in the left panel of Figure 3.

3. **Contextual Entropy $\sim$ Cross-Entropy Loss**: Equation 4 shows that the contextual entropy is the expected value of the surprisal, with the expectation computed using the next token probabilities $p_\theta$. When we consider a large number of tokens in the prompts, we expect the contextual entropy to be almost equal to the surprisal of the next token predictions when averaged over all the tokens. This can be seen empirically in the right panel of Figure 3.

**Why are logit IDs and contextual entropy correlated?** Given that the second step of the relation above is non-trivial, we dedicate a more in-depth study. We wish to demonstrate that the correlation between logit IDs and the contextual entropy (entropy of next token predictions) is a fundamental property of the softmax layer, making this relationship more general.

Given $\mathbf{z} = (z_1, z_2, ... z_{|\mathcal{V}|})$ where $z_\alpha \in \mathbb{R}$, as the input to a softmax layer, the associated entropy of the probability distribution generated by the softmax operation is

$$S(\mathbf{z}) = - \sum_{\alpha=1}^{|\mathcal{V}|} p(\mathbf{z})_\alpha \log p(\mathbf{z})_\alpha = \left( \log \sum_{\alpha=1}^{|\mathcal{V}|} e^{z_\alpha} - \frac{\sum_{\alpha=1}^{|\mathcal{V}|} z_\alpha e^{z_\alpha}}{\sum_{\alpha=1}^{|\mathcal{V}|} e^{z_\alpha}} \right), \quad (6)$$

where $p(\mathbf{z})_\alpha = \dfrac{e^{z_\alpha}}{\sum_{\beta=1}^{|\mathcal{V}|} e^{z_\beta}}$ is the probability of the $\alpha^{\text{th}}$ word in a vocabulary with $|\mathcal{V}|$ entries.

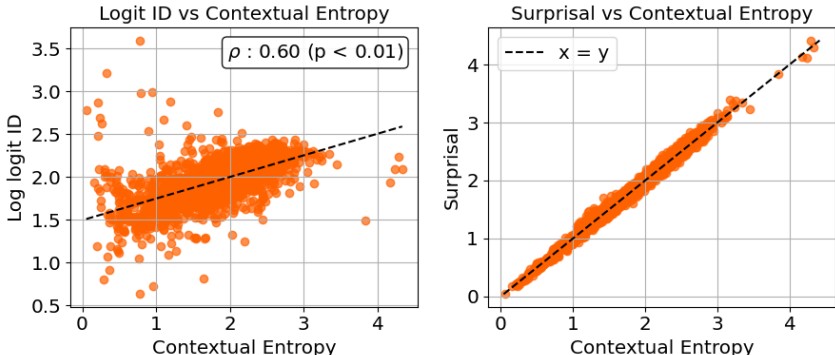

Figure 3: **Correlating intrinsic dimension at the last layer to surprisal.** The points in the following plots are calculated using the 2244 prompts considered in this paper for the LLAMA model. - (a) Left Panel: analysis of the correlation between the logits ID to the average contextual entropy and (b) Right Panel: comparing the average contextual entropy to the average surprisal. On the left panel, we observe a correlation of $\rho = 0.6$ between the logits ID and contextual entropy, and on the right panel, we notice that the surprisal is approximately equal to the contextual entropy. The intrinsic dimension was calculated at scaling $= 4$, refer to Figure 14 for scaling = 2 and 8.

When the next token predictions are obtained using the softmax activation function, the contextual entropy reduces to the above expression, which we refer to as the softmax entropy[5]. From Equations 5 and 6, we see that the average contextual entropy for a prompt $X$ is the average of the softmax entropy of the corresponding logits -

$$\mathcal{H}(X) = \frac{1}{N} \sum_{i=1}^{N} S(\mathbf{z}_i). \tag{7}$$

These relations suggest that the underlying manifold on which the logits lie plays a role in the evaluation of the entropy. Given this manifold $\mathcal{M}$ with measure $\mu$, and the logits $\mathbf{z}$ distributed according to the density function $P(\mathbf{z})$, the expected value of the softmax entropy is given by

$$\langle S \rangle_\mathcal{M} = \int_\mathcal{M} d\mu(\mathbf{z}) \, P(\mathbf{z}) S(\mathbf{z}). \tag{8}$$

From what is observed empirically, we expect that the dimension of the manifold $\mathcal{D}_\mathcal{M}$, typically much smaller than $|\mathcal{V}|$, should play a role in the integral.

We can show this explicitly in a toy example. Let us suppose that the next-token probabilities are uniformly distributed over a probability simplex $\Delta_{\mathcal{D}_\mathcal{M}}$. In this case, $P(\mathbf{z}) \in \Delta_{\mathcal{D}_\mathcal{M}}$ is drawn from the Dirichlet distribution with $\boldsymbol{\alpha} = 1$, where the expected entropy (Wolpert & Wolf, 1995; Nemenman et al., 2001) is given by

$$\langle S \rangle_{\Delta_{\mathcal{D}_\mathcal{M}}} = \psi(\mathcal{D}_\mathcal{M} + 1) - \psi(2) = \sum_{k=1}^{\mathcal{D}_\mathcal{M}} \frac{1}{k} - 1 \tag{9}$$

where $\psi$ is the Digamma function. From the above relation and using bounds on the harmonic number (Viola, 2017), it can be shown that

$$\left( \log \mathcal{D}_\mathcal{M} - \frac{1}{2} \right) < \langle S \rangle_{\Delta_{\mathcal{D}_\mathcal{M}}} \leq \log \mathcal{D}_\mathcal{M} \tag{10}$$

and in the asymptotic limit,

$$\lim_{\mathcal{D}_\mathcal{M} \to \infty} \langle S \rangle_{\Delta_{\mathcal{D}_\mathcal{M}}} = \log \mathcal{D}_\mathcal{M} + \gamma - 1 \sim \log \mathcal{D}_\mathcal{M} - 0.42 \tag{11}$$

where $\gamma$ is the Euler-Mascheroni constant. We thus observe a $\log \mathcal{D}_\mathcal{M}$ dependence for the expected entropy in the probability simplex $\Delta_{\mathcal{D}_\mathcal{M}}$. While this relation might not hold for a generic manifold, it would be worth investigating this in more detail.

---

[5]We use $S(\mathbf{z})$ to denote the softmax entropy that is defined at the level of logits and $H(x_{<i})$ to denote the contextual entropy which is more generically defined at the level of tokens. For clarity, we use the Greek letters to indicate the index in vocabulary and the Roman letters to indicate indices of tokens in a prompt.

**Relation to previous work.** The connection between the surprisal and ID was discussed in (Cheng et al., 2023) where the correlation was calculated between the peak ID of the dataset of the last token representations and the log of dataset perplexity in Fig. 2 of Cheng et al. (2023). However, we get a correlation in similar spirit at a finer level since it reveals a correlation at the level of individual prompts (more detailed comparison in Sec. D).

## 6 CASE STUDY: PRE-OUTPUT DETECTION OF MALICIOUS PROMPTS VIA PROMPT GEOMETRY

The analyses in Sections 4 and 5 indicate that prompt-level intrinsic dimension encodes meaningful information about how a model organizes and resolves uncertainty across layers. This suggests a generic pre-output diagnostic: if different classes of prompts drive the model into measurably different geometric regimes, then the layerwise ID profile should provide a compact feature for flagging such prompts before decoding, without task-specific heuristics or access to generated text. We therefore ask whether a simple linear probe on ID profiles can distinguish benign from malicious prompts, as a proof-of-principle that the geometric signal is actionable.

**Datasets description.** This paper uses different curated datasets for jailbreak detection, taken from three distinct public sources on HuggingFace: i) a dataset of malicious prompts[6], ii) a dataset of jailbreak attempts and benign prompts [7], and iii) a dataset containing exclusively jailbreaking prompts[8], from which we consider the "attack" prompts. To create a balanced collection from this source, we supplemented it with benign text samples extracted from the Pile dataset utilized in the previous Sections. For clarity, we call these three datasets MALICIOUS, JAILBREAK and ATTACK, respectively. Finally, to standardize the data for our experiments, we filter the combined collection, retaining only those prompts with a token count in the range of $[500, 1000]$, and considering subsets of 3000 jailbreak samples and 3000 benign samples.

**Experiment description.** We process each benign/malicious prompt of a given dataset through LLAMA and compute ID at each layer. [9] With the ID profile data vector, we train a linear classifier (logistic regression) to predict a binary label (benign vs malicious) with a split of 80-20 between training and test set. We compare results to the following alternatives:

- **Guardrail Models**: We use two LLMs trained as safety classifiers. Llama Guard 8B (Inan et al., 2023) performs a fine-grained classification, identifying the specific category of policy violation from its safety taxonomy. Shield Gemma 9B (Zeng et al., 2024) offers a binary safe/unsafe classification based on a set of guidelines, all of which were used in our experiments. Both models are configured to operate exclusively on the input prompt.

- **TunedLens entropy**: The Tuned Lens (Belrose et al., 2023) analyzes transformers from the perspective of iterative inference (Jastrzebski et al., 2018), understanding how model predictions are refined layer by layer. This is done by training linear probes to decode intermediate hidden states into the model's vocabulary space. Using this, we analyze the hidden states by computing the average entropy of the unembedded predictions across all the tokens in a prompt. We provide details on how this is computed in Appendix F.

We quote results in Table 1 for LLAMA. A linear classifier on the ID profile achieves 90–95% accuracy across datasets and models, substantially outperforming Llama Guard and Gemma Shield (60–70% on the same splits). TunedLens entropy features attain similar performance (90–95%). We argue that this is consistent with the geometry–uncertainty link in Section 5, and investigate the correlation between ID and the entropy of latent predictions in Appendix F.

---

[6]https://huggingface.co/datasets/guychuk/benign-malicious-prompt-classification

[7]https://huggingface.co/datasets/Bogdan01m/Catch_the_prompt_injection_or_jailbreak_or_benign

[8]https://huggingface.co/datasets/Bravansky/compact-jailbreaks

[9]Here we quote results for GRIDE only since results on estimators TLE and ESS are quantitatively similar.

| Dataset | Shield Gemma | Llama Guard 3 | TunedLens Entropy | GRIDE |
|---|---|---|---|---|
| MALICIOUS | 0.58 | 0.64 | 0.96 | 0.94 |
| JAILBREAK | 0.52 | 0.53 | 0.65 | 0.92 |
| ATTACK | 0.67 | 0.80 | 0.98 | 0.90 |

Table 1: **Cross-validation accuracy from logistic regression for pre-output safety screening.** Results for the binary classification of malicious/benign prompts processed through LLAMA from three different datasets. Values indicate cross-validation accuracy run on 5 different train/test splits with 80/20 ratios. All uncertainties are within subpercent.

## 7 CONCLUSIONS

We model prompts as token clouds and use intrinsic dimension to probe their empirical measure across layers, showing that it peaks in early–middle layers, increases under semantic disruption, and aligns with next-token uncertainty via a logits–softmax link. This yields a compact, interpretable, pre-output signal: a linear probe on the layerwise ID profile can flag malicious prompts, requiring no decoding. Conceptually, this positions geometry as a bridge between mechanistic tools of iterative inference and practical safety interventions.

In this work, we demonstrated that prompt-level intrinsic dimension can effectively detect malicious prompts, achieving 90-95% accuracy and substantially outperforming existing safety tools. This success suggests that prompt geometry holds promise for a broader range of practical applications beyond safety screening. The geometric signatures we identified could potentially be leveraged for tasks such as detecting hallucinations, identifying out-of-distribution inputs, or predicting generation quality before decoding. Additionally, while we focused on intrinsic dimension as our geometric observable, exploring other geometric properties such as curvature, clustering metrics, or topological features could reveal complementary signals about how models process different types of content. These geometric approaches offer an unsupervised, interpretable window into model behavior that operates directly on internal representations without requiring labeled data or task-specific fine-tuning. The strong correlation we established between geometry and uncertainty further suggests that monitoring these geometric properties during inference could enable real-time interventions, potentially leading to more controllable language models.

**Limitations.** Our analysis relies on kNN-based estimators that assume local uniformity, and the ID estimates are reliable when the prompts are sufficiently long ($\mathcal{O}(500)$ tokens), as seen in Section 6, limiting the applicability of pre-output screening. Moreover, absolute ID values can vary with estimator choice and neighborhood scale, and a more comprehensive multiscale analysis is reserved for future work. The theoretical connection between manifold geometry and entropy is demonstrated in toy settings and empirically; a general proof for realistic logit manifolds remains open.

## 8 REPRODUCIBILITY

All the results contained in this work are reproducible by means of an anonymised repository that can be found at this https://anonymous.4open.science/r/token_geometry-9DBC/.

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

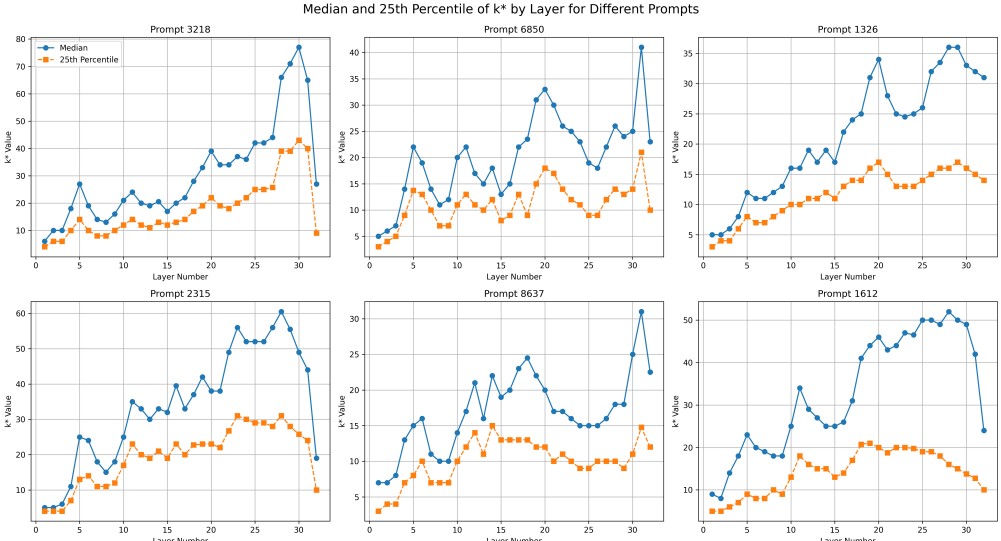

Figure 4: **Check for local uniform density for tokens in a prompt across layers.** The plots correspond to the local homogeneity check for 6 prompts across the dataset for representation obtained using LLAMA. In each plot, we find the median and the $25^{\text{th}}$ quantile of $k^*$ across tokens for a given prompt. Here $k^*$ stands for the neighborhood until which the local density aprroximation holds. We do this for every layer given by the $x$-axis.

## A    MORE DETAILS ON INTRINSIC DIMENSION ESTIMATORS

**Local uniform density check.**    We note that the above kNN-based estimators rely on the assumption of local homogeneity, that is, nearby points are assumed to be uniformly sampled from d-dimensional balls. As a check of local uniform density, we employ the Point Adaptive kNN (PAk) method introduced in Rodriguez et al. (2018). PAk determines the extent of the neighborhood over which the probability density can be considered constant for each data point, given a predefined confidence level. We show the results in Figure 4, where we plot the layerwise median (and $25^{\text{th}}$ quantile) values of $k^*$, the maximum $k$ (neighborhood) around a token where the density can be assumed to be constant. This implies the density around tokens is constant on average until large values of k $\sim 20$.

**Pareto distribution of log ratios.**    To test the validity of GRIDE, we provide evidence that token representations meet a necessary condition for this assumption to hold in the context of the TwoNN estimator.

In Facco et al. (2017), where the TwoNN estimator is introduced, the authors show that local homogeneity leads to a linear relationship between quantities related to the log-ratios and the empirical cumulative distribution. In particular, the linear relationship is between quantities related to the log-ratios $\left(\mu = \dfrac{r_2}{r_1}\right)$, and its empirical cumulative distribution. The linear relationship is between $\log \mu$ and $-\log\left(1 - F^{emp}\left(\mu_i\right)\right)$. For a more elaborate explanation, please refer to the section "A Two Nearest Neighbors estimator for intrinsic dimension" in Facco et al. (2017). The intrinsic dimension is then estimated as the slope of this line, as illustrated in Figure 1 of Facco et al. (2017).

We check if this distribution results in a straight line in our case by performing this check on prompt 3218 for all layers in figure 5. It can be seen that this results in a distribution implying that the token representations satisfy a necessary condition of the local homogeneity hypothesis.

The plot uses neighbors with $k = 1$ and 2, as required by the TwoNN estimator. A natural next step is to evaluate the distribution using $k = 2$ and 4, consistent with our range scaling factor of 4. However, this results in a distribution that is not linear, suggesting that a more sophisticated test is needed.

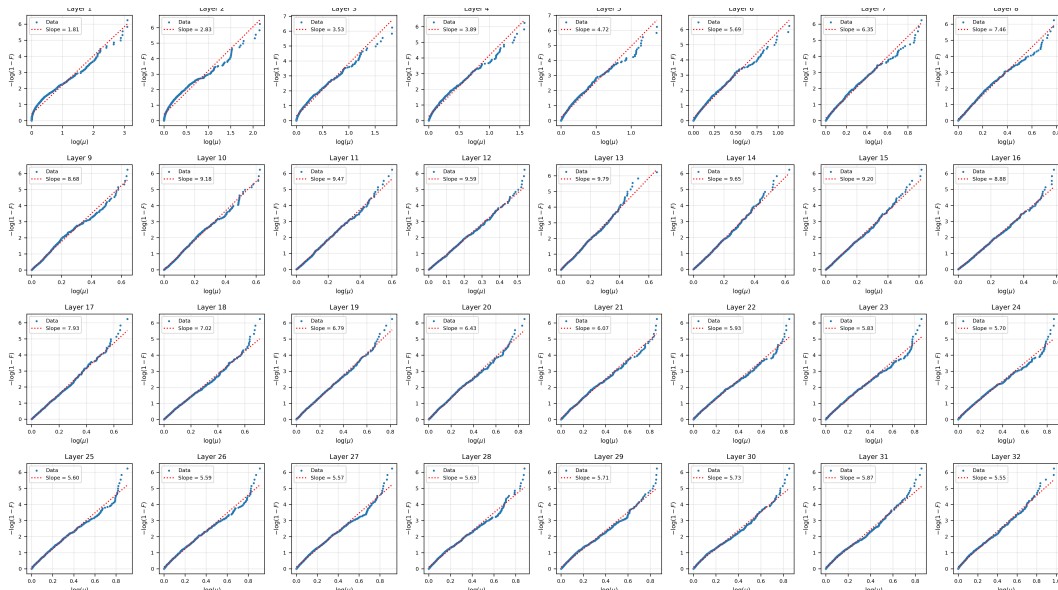

Figure 5: **Local homogeneity across layers.** Empirical check of the TwoNN assumption for prompt 3218 using representations extracted from LLAMA. The plot shows the relationship between $\log \mu$ and $-\log(1 - F^{emp}(\mu_i))$ across all layers, indicating that token representations satisfy the Pareto condition of the log ratio distances.

**Range scaling analysis for GRIDE.** While in the main text we always use range scaling $= 4$, here we show the stability as a function of scaling. The prompts we analyze have $N = 1024$ tokens and in Fig. 6 we check the dependence of ID estimate on range scaling $\in \{2, 4, 8, ..256\}$ for prompt 3218, which suggests the choice of scaling $= 4$.

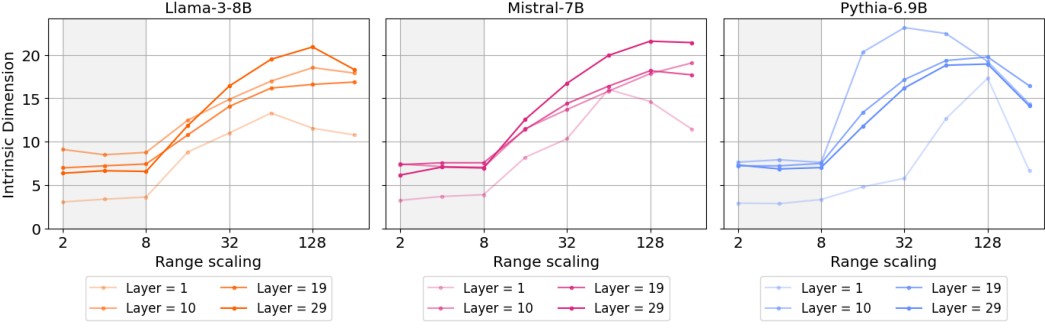

Figure 6: GRIDE scale analysis for an unshuffled prompt (prompt number 3218) across layers.

## B    DETAILS ON THE SHUFFLING METHODOLOGY

The shuffling algorithm explained in the main text is schematically summarized in Figure 7, where we also include a toy example. In this example, we have $\hat{S} = 2$ since we consider 16 tokens, whereas in the experiments, we have $\hat{S} = 5$ because we have $1024 = 4^5$ tokens.

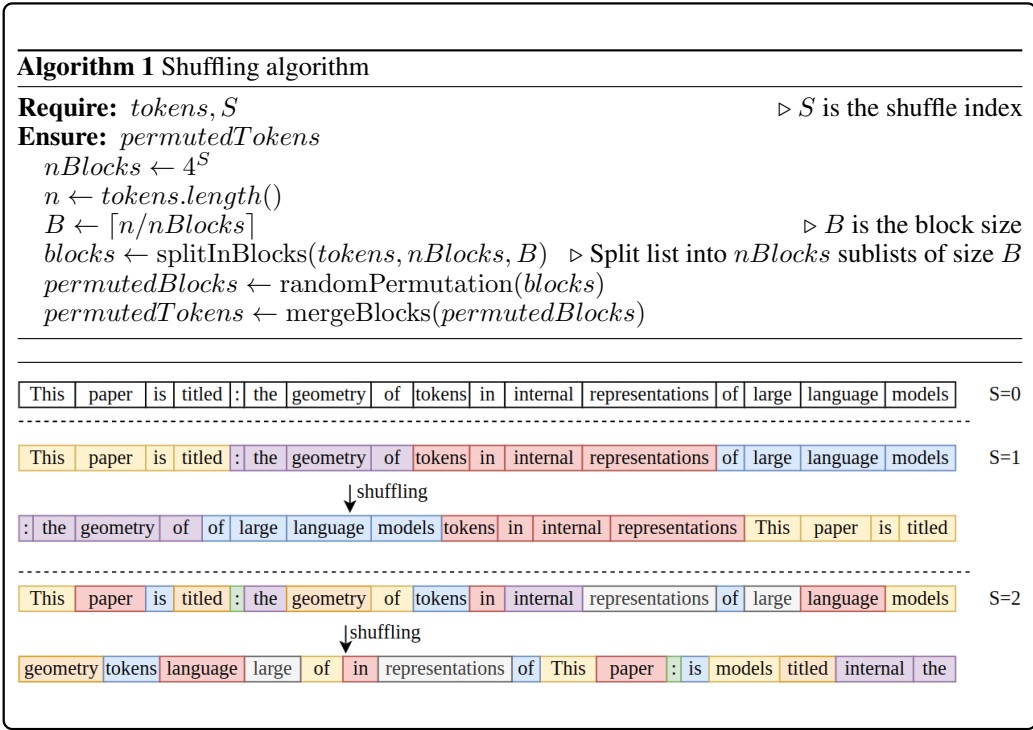

**Algorithm 1** Shuffling algorithm

**Require:** $tokens, S$          ▷ $S$ is the shuffle index
**Ensure:** $permutedTokens$
    $nBlocks \leftarrow 4^S$
    $n \leftarrow tokens.length()$
    $B \leftarrow \lceil n/nBlocks \rceil$          ▷ $B$ is the block size
    $blocks \leftarrow \text{splitInBlocks}(tokens, nBlocks, B)$    ▷ Split list into $nBlocks$ sublists of size $B$
    $permutedBlocks \leftarrow \text{randomPermutation}(blocks)$
    $permutedTokens \leftarrow \text{mergeBlocks}(permutedBlocks)$

Figure 7: **The shuffling algorithm with an example.** Top Panel: Algorithmic description of the shuffling procedure described in Section 4. Bottom Panel: An example of the shuffling algorithm using $N = 16$ tokens. The first row ($S = 0$) corresponds to the unshuffled sequence. When $S = 1$, the tokens are split into $4^1$ blocks first and then, the blocks are shuffled. The last row $S = 2$ shows the fully shuffled case where the tokens are randomly permuted.

## C    CONSISTENCY OF RESULTS FROM THE SHUFFLE EXPERIMENT

In this section, we show the consistency of the results that were discussed in Section 4. In Figure 8 we show 6 shuffled and unshuffled random prompts, computed for three different models: LLAMA, MISTRAL and PYTHIA, showing qualitatively similar behaviour.

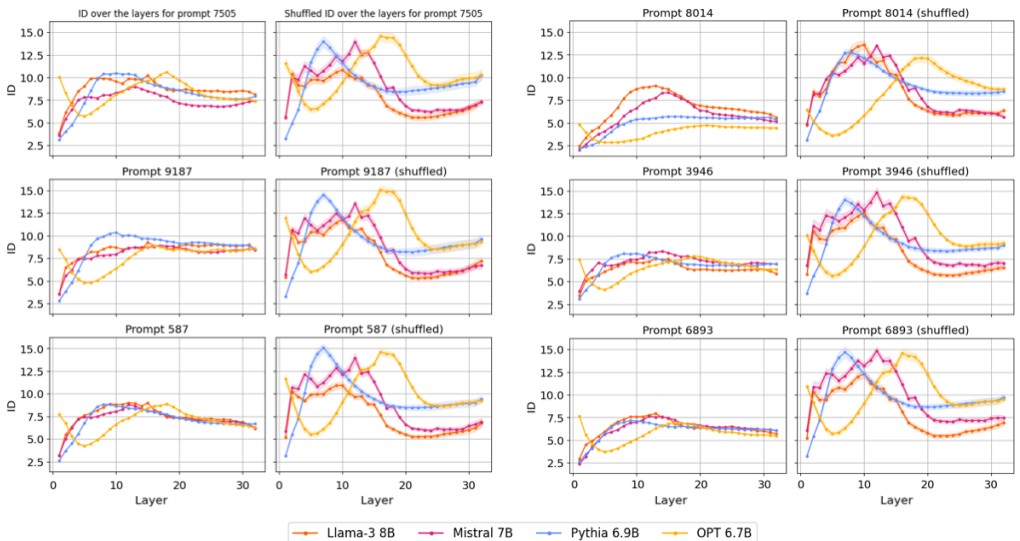

Figure 8: The curves have been calculated for 6 random prompts from the Pile-10K dataset at scaling = 4 where the title indicates the prompt number in the dataset. Left Panels: intrinsic dimension for a single prompt as a function of layers for different models, including the embedding layer. Right Panels: intrinsic dimension for the shuffled version of the same prompt averaged over 20 random permutations of tokens. The shaded regions indicate the standard deviation from the mean.

We also want to verify that our main conclusions are robust to the choice of intrinsic dimension estimator. Using ESS, TLE, and GRIDE, we replicate the shuffle experiment finding consistently behaviour across estimators, see Figure 9.

# D  QUALITATIVE COMPARISON OF PROMPT-LEVEL GEOMETRY TO PREVIOUS DATASET-LEVEL STUDIES

Previous work (Ansuini et al., 2019; Doimo et al., 2020; Pope et al., 2021; Valeriani et al., 2023; Cheng et al., 2023; 2024; Antonello & Cheng, 2024) have studied internal representations from a geometric point of view by considering point clouds of last token representations. While the approach is similar in spirit, prompt-level and dataset-level measures of intrinsic dimension probe different manifolds and thus different features of LLMs. We expect this because the relationship between the last tokens is different from that of tokens within the same prompt. In the upcoming analysis, we understand this difference intuitively by looking at the geometry of the shuffled and the unshuffled prompts at the dataset and prompt-level around the peak layers.

While dataset-level and prompt-level ID profiles exhibit similar behavior qualitatively, e.g. they peak in early-middle layers, there is a notable difference in the shuffled and unshuffled prompts. At the dataset level, we see that the unshuffled ID has a more prominent peak than the shuffled ID, whereas it is the other way around at the prompt level. In the shuffled case, the last token representations are less likely to share semantic content, leading to a lower intrinsic dimension at the dataset level. At the prompt level, the lesser prominence of the peak of the unshuffled case can be explained using the ID loss correlation. Since the loss is expected to be lower for the unshuffled prompts, we can expect their ID peak to be less prominent than that of the shuffled prompts.

For the dataset-level analysis, we use a corpus of $2244$ prompts (the same corpus used for the prompt-level analysis), drawn from Pile-10K and consisting of prompts with at least $1024$ tokens. The last token representations are extracted from these prompts as follows - we choose tokens at positions $512$ through $532$ that result in a $20$-token sequence for the unshuffled case[10]. We randomly permute aforementioned the $20$-token sequences in the shuffled case and obtain the last token representations.

---

[10]This is a simplified setup of the experiments in Cheng et al. (2024).

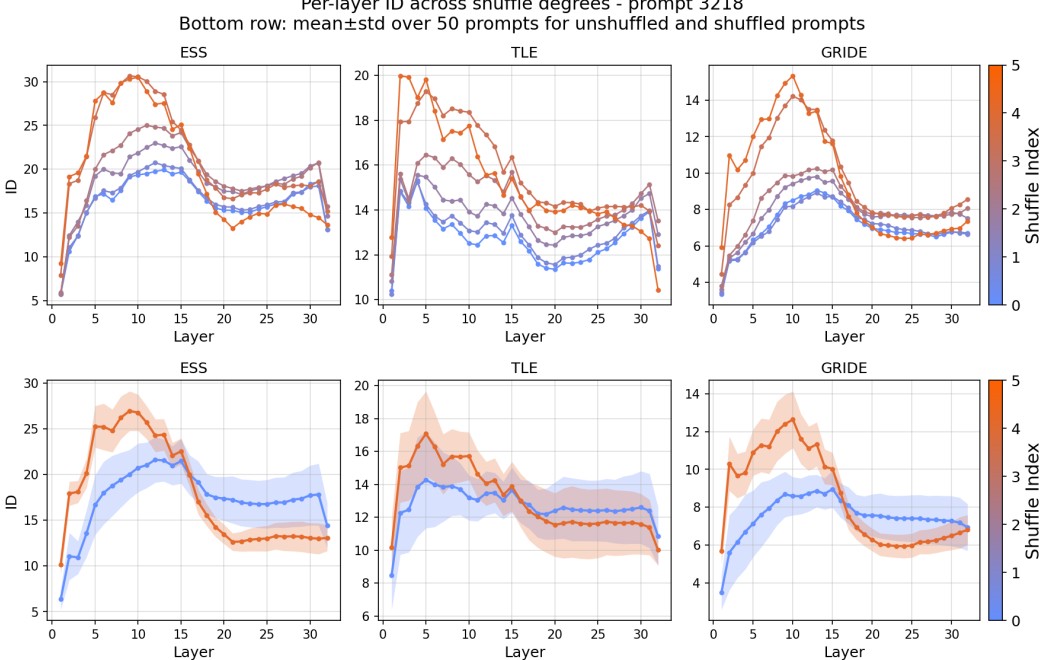

Figure 9: **Shuffle experiment using different estimators.** The qualitative trend from section 4 holds across all intrinsic-dimension estimators—ESS ($k$=10), TLE ($k$=20), and GRIDE (range scaling = 4): the estimated ID increases with the shuffle index $S$. *Top row:* intrinsic dimension for a single random prompt as a function of model layers (color bar indicates shuffle index $S$). *Bottom row:* intrinsic dimension averaged over all $2244$ prompts as a function of layers for the structured case ($S$=0) and the full shuffle ($S$=5); shaded bands denote $\pm 1$ standard deviation. All curves are computed for the LLAMA model.

The prompt-level analysis is done on prompt number $3218$ from the Pile-10K dataset. In Figure 10, we plot the t-SNE projections of the shuffled and unshuffled along with ID for different scalings at both the dataset and prompt levels. We notice that in both levels, the shuffled and unshuffled representations lie on separate manifolds (Sarfati et al., 2024).

For the sake of completeness, we compare the results of the ID-loss correlation at the dataset and the prompt level in the next section.

## D.1 PROMPT LEVEL ID IS MORE STRONGLY CORRELATED TO SURPRISAL

Since there is an extensive amount of work done for the case of Opt-6.7B at the dataset level regarding the ID-surprisal correlation, we compare the prompt level results to the dataset level for Opt-6.7B. Before proceeding here is a summary of the dataset level results from Cheng et al. (2023) and Cheng et al. (2024) that are relevant for our comparison.

- In Cheng et al. (2023), the authors show a positive Spearman correlation of $0.51$ for Opt-6.7B (Figure 2a in Cheng et al. (2023)) using the ID estimator Expected Simplex Skewness (ESS) (Johnsson et al., 2015) between ID at the peak and the surprisal.

- An analysis at a higher range scaling is done in Cheng et al. (2024) where they show a **negative correlation** with surprisal (Figure 6 in Cheng et al. (2024)) among a population consisting of different models and datasets with a relatively less statistical significance since it has a high $p$-value = $0.09$.

On the other hand, using the prompt-level approach, we measure a **higher layerwise positive correlation** with surprisal. We summarize the results in Table 2.

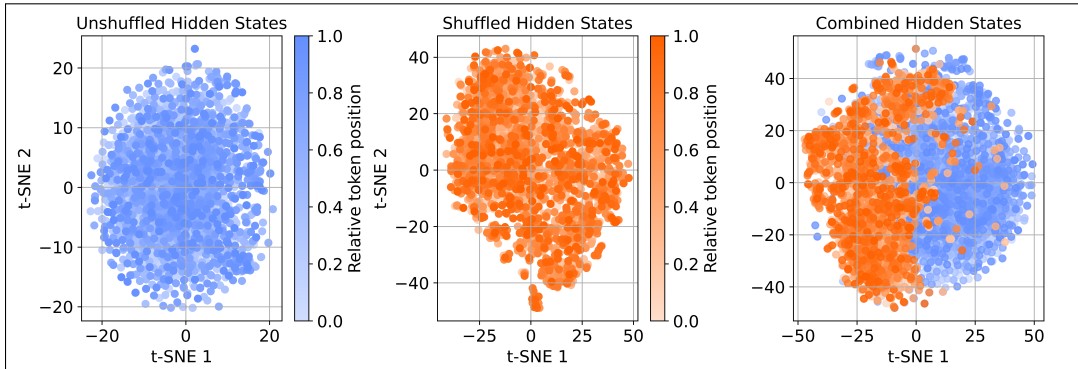

(a) **Dataset level.** Left and middle panels: The t-SNE plots represent the 2244 last token representations for both the unshuffled and the shuffled cases. Right panel: The combined t-SNE projection comprises 4488 last token representations from both cases.

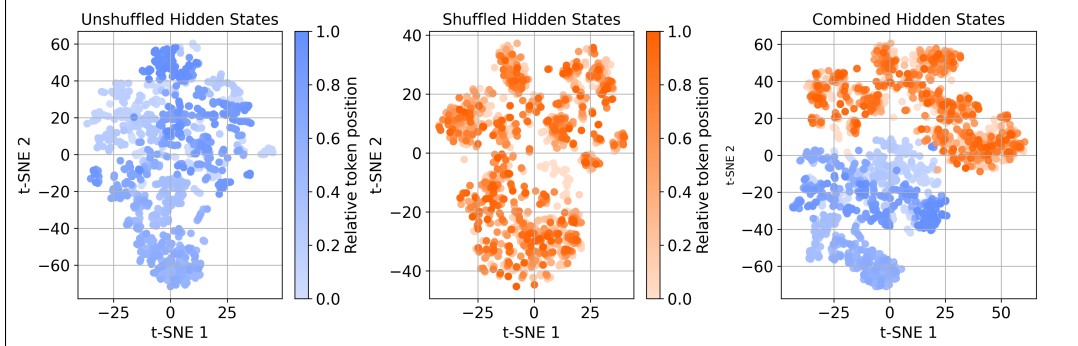

(b) **Prompt level.** Left and middle panels: The t-SNE projections of token representations for prompt number 3218 from Pile-10K, truncated to 1024 tokens, for both the unshuffled and shuffled cases. Right panel: The combined t-SNE projection comprises 2048 token representations from both cases.

Figure 10: **Dataset geometry and Prompt geometry.** A qualitative comparison of last-token representations at the dataset level (top panel) and the prompt level (bottom panel) geometry at layer 11 using t-SNE projections. All the plots are obtained using the representations from LLAMA.

|  | Dataset level (ESS) | Dataset level (2NN) | Dataset level (high scaling) (many models × corpus) | Prompt level (2NN) | Prompt level (scaling = 8) |
|---|---|---|---|---|---|
| Spearman $\rho$ | 0.51 | 0.13 | -0.46 | 0.69 | 0.73 |
| $p$-value | 0.01 | 0.5 | 0.09 | $< 0.01$ | $< 0.01$ |

Table 2: Summary of Spearman correlations between ID and loss from prompt and prompt level analysis for Opt-6.7B. The results for prompt level are from Figure 11 and the dataset level are from Cheng et al. (2023) and Cheng et al. (2024).

# E    CONSISTENCY CHECKS FOR THE CORRELATION BETWEEN INTRINSIC DIMENSION AND LOSS

In this section we provide further checks to support the results of the main text. In Figure 12, we show that the correlation between $\log$ ID and surprisal is present also at range scalings 2 and 8 for the GRIDE estimator, though weaker for range scaling 2. In Figure 13 we show that the correlation between the intrinsic dimension of adjacent layers and the last layer is strong, providing a link between ID of internal layers with surprisal.

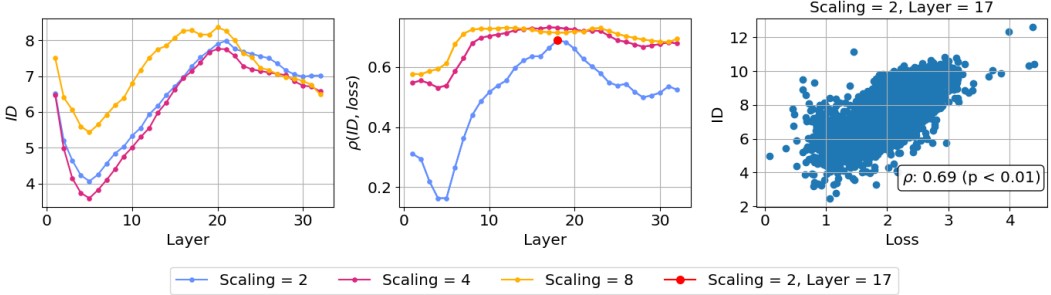

Figure 11: **Summary of results for Opt-6.7B at the prompt-level.** Left panel: The ID curve for Opt-6.7B for scaling = 2, 4, 8 for prompt number 3218 from Pile-10K. We observe a peak around layer 20 as in the dataset level (Cheng et al., 2024). Middle panel: Spearman correlation between ID and loss for Opt-6.7B for different range scalings at the prompt level as a function of layers. Right panel: Scatter plot with the $ID$ (y-axis) and the average surprisal (x-axis) at scaling = 2, layer 17 for the 2244 prompts we consider in this text.

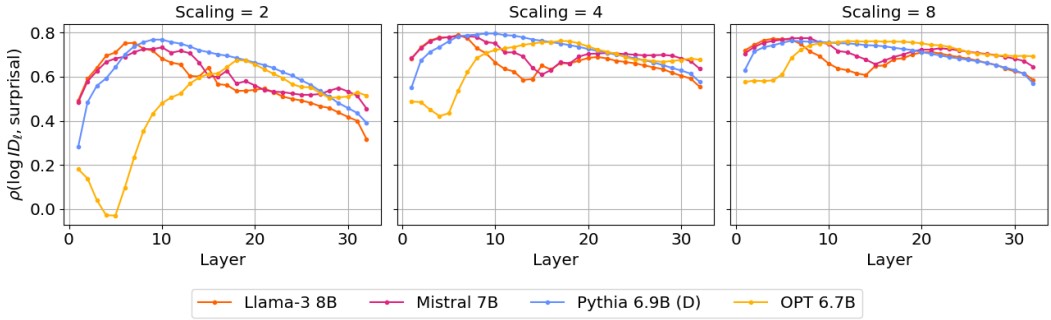

Figure 12: **Scale analysis for the correlation between intrinsic dimension and loss.** Pearson coefficient between the logarithm of intrinsic dimension and model loss at scalings = 2, 4, 8 for different models.

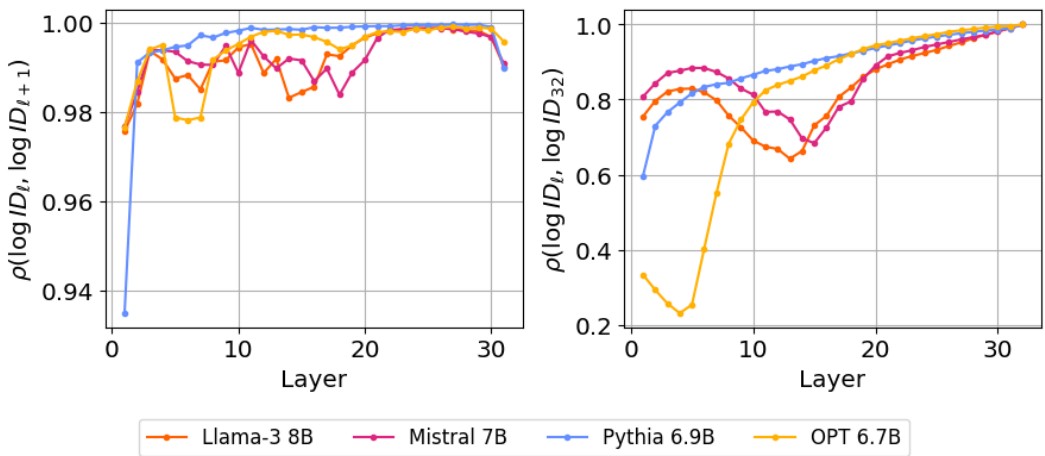

Figure 13: **Correlation between the intrinsic dimension of adjacent layers and the last layer.** Pearson coefficient between the log of intrinsic dimension at layer $\ell$ and $\ell + 1$ (left panel) and between layer $\ell$ and the last layer $\ell = 32$ for different models as a function of layers. The four curves correspond to LLAMA (orange), MISTRAL (magenta), PYTHIA (blue), and OPT (yellow). The $p$-values for the Pearson coefficients in this plot are below $0.01$.

In Figure 14, we extend the analysis in Figure 3 to other range scalings by checking the scatter plot between the logits ID and the contextual entropy.

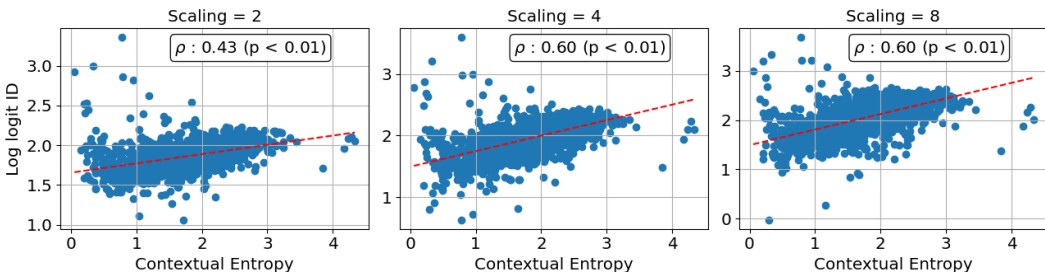

Figure 14: **Scale analysis for the correlation between intrinsic dimension of logits and contextual entropy.** Pearson coefficient between the logarithm of the intrinsic dimension of the logits and model contextual entropy for scalings $= 2, 4, 8$ for LLAMA.

## F  THE CORRELATION BETWEEN INTRINSIC DIMENSION AND ENTROPY OF THE LATENT PREDICTIONS

In section 5, we discuss a correlation between the ID of the internal representations across the layers and the surprisal of the model's next token predictions. We extend the analysis to study how intrinsic dimension relates to the statistical properties of the latent predictions in this section. Since the latent predictions are obtained by unembedding the hidden states (nostalgebraist, 2020), we expect the statistical properties of the latent predictions to be related to the geometric properties of the hidden states. This motivates the analysis of the relation between intrinsic dimension of the prompts and the average entropy of the latent predictions obtained from TunedLens (Belrose et al., 2023).

For each layer $\ell$, the TunedLens consists of learning an affine transformation of a hidden state $(x(\ell) \in \mathbb{R}^d)$ so that its image under unembedding is as close to the output logits as possible. This is implemented by training translators $A_\ell \in \mathbb{R}^{d \times d}, b_\ell \in \mathbb{R}^d$ to minimize the following loss -

$$\min_{A_\ell, b_\ell} \mathcal{D}_{KL} \left( f_{>\ell} \left( x(\ell) \right) \| \text{LayerNorm} \left( A_\ell x(\ell) + b_\ell \right) W_U \right) \tag{12}$$

where $f_{>\ell} \left( x(\ell) \right)$ is the model's output logits corresponding to the hidden state $x(\ell)$ and $W_U$ is the unembedding matrix. By doing so, the TunedLens finds a latent prediction corresponding to a hidden state $x(\ell)$ given by

$$q_\ell(v|x(\ell)) = \text{softmax}( \text{LayerNorm} \left( A_\ell x(\ell) + b_\ell \right) W_U) \tag{13}$$

where $q_\ell(v|x(\ell))$ is a probability distribution over the model's vocabulary $\mathcal{V}$. Note that the latent predictions are obtained in a manner similar to the model's prediction from the last layer representations, with the key difference being the application of affine translators.

Given a prompt $X$ consisting of tokens represented as $(x_1(\ell), x_2(\ell), ...x_N(\ell))$ at layer $\ell$, we calculate the average entropy of the latent predictions at layer $\ell$

$$H_\ell(X) = -\frac{1}{N} \sum_{i=1}^{N} \sum_{v \in \mathcal{V}} q_\ell \left( v \mid x_i(\ell) \right) \log q_\ell \left( v \mid x_i(\ell) \right) \tag{14}$$

We plot the Pearson correlation between the intrinsic dimension and the entropy of the latent predictions ($H_\ell$) on the population of 2244 prompts across different models in Figure 15. Since the latent predictions in the later layers are closer to the model's prediction, we can expect its correlation with the intrinsic dimension to be similar to the trend in Figure 2. We notice that in both cases (Figures 2, 15), $\rho > 0.5$ for LLAMA, PYTHIA, and OPT in the last layer consistent with our expectation. We summarize the correlation between latent entropy and intrinsic dimension seen in Figure 15 as follows -

1. GPT-2 models (left panel): There is a notable positive correlation $\rho > 0.5$ between the two quantities from the early layers onwards. This implies that the prompts that have a high dimensional representation tend to have a higher latent prediction entropy.

2. PYTHIA models (middle panel): In this case, we observe a positive correlation $\rho > 0.5$ from the middle layer onwards. There is a similar trend in the correlation across different model sizes.

3. LLAMA and OPT (right panel): In these models, we do not see a positive correlation until the late layers. In LLAMA, we see a moderate negative correlation ($\rho \sim -0.5$) around layer 20.

In the GPT-2 and PYTHIA models, we notice a positive correlation from the middle layers onwards according to our expectations. However, the negative correlation found in LLAMA requires further understanding, which we leave for future work.

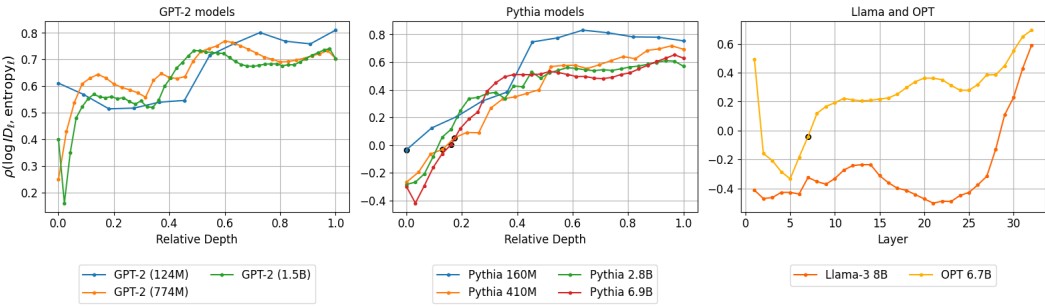

Figure 15: **Correlation between intrinsic dimension and the layerwise entropy from TunedLens.** Pearson coefficient between the logarithm of the intrinsic dimension and the entropy of the latent predictions for different models as a function of layers. Left panel: GPT-2 models consisting of GPT-2 (blue), GPT-2 Large (orange) and GPT-2 XL (green). Middle panel: PYTHIA (deduplicated) models with 160M (blue), 410M (orange), 2.8B (green) and 6.9B (red) parameters. Right panel: LLAMA (orange) and OPT (yellow). The black marker indicates the $p$-values that are above $0.01$. We notice a consistent positive between the latent prediction entropy and the intrinsic dimension in the GPT-2 models, a positive correlation from the middle layers for the PYTHIA models and a negative correlation until the late layers for LLAMA.

