# OpenReview forum: "The Intrinsic Dimension of Prompts in Internal Representations of Large Language Models"
_ICLR.cc/2026/Conference — Submitted to ICLR 2026_

### Official Review · Reviewer_8ot4 · 2025-10-20

**Soundness:** 2
**Presentation:** 2
**Contribution:** 2
**Rating:** 2
**Confidence:** 4

**Summary:**

The manuscript studies the intrinsic dimension of prompts, modeled as clouds of tokens, in the embedding spaces of large language models. They present a handful of empirical findings; the intrinsic dimension of prompts appears to be correlated to surprisal and increases when prompts are shuffled, for example. They show that layerwise intrinsic dimension features can be used to identify malicious inputs.

**Strengths:**

The paper is fairly easy to understand. Discussion of related work is well-researched and thorough. The correlations between ID and surprisal seem to be pretty robust, as is the "peak" observed in shuffling experiments, and I appreciate that the authors used diverse prompts from the Pile as opposed to ones from some more narrow domain.

**Weaknesses:**

There are a few things missing from the paper that I think would strengthen it greatly:

1. Efficiency analysis: I'm not familiar with the GRIDE algorithm. How computationally intensive is it compared to the other baselines?
2. The main applied results are all from one model. Given the differences observed between models elsewhere (e.g. in Appendix F), I think there's a good chance the others would perform differently here as well.
3. The comparison to Shield Gemma and Llama Guard aren't exactly fair, since those are zero-shot methods. I'd like to see more baselines tuned on this particular distribution (e.g. finetuned BERT)
4. The fact that this method is only evaluated on prompts of a specific length is a significant omission; the authors need to include results for prompts of different lengths and results averaged across all lengths.
5. You can't claim that ID has value as a "pre-output signal" without running pre-output experiments. How do the results change when you only run the first k layers of the model under evaluation?

**Questions:**

Why is Tuned Lens so much worse on jailbreak prompts compared to "malicious" and "attack" prompts? This is not mentioned in the text, but only in Table 1. Is the result a typo?

How exactly is the Tuned Lens baseline computed? Are you using the full vector of entropies at each layer as inputs to the classifier?

Do you observe the "peak" from Figure 1 on malicious prompts, or just shuffled ones?

---

> ### Author Response · Authors · 2025-11-21
> **Reply to reviewer 8ot4**
>
> We thank the reviewer for a careful review of our manuscript and their constructive feedback on weaknesses. We address each point below.
>
> Weaknesses
>
> > Efficiency analysis: I'm not familiar with the GRIDE algorithm. How computationally intensive is it compared to the other baselines?
>
> We agree with the reviewer that this is an important point. We estimated an average of ~30 samples per second for GRIDE, ~3 samples per second for ESS and ~1 sample per second for TLE. Samples here refer to single point clouds (i.e. prompts.). This refers to the time required for the estimators to compute ID after the distance matrix has been computed, which is the computational bottleneck.
>
> > The main applied results are all from one model. Given the differences observed between models elsewhere (e.g. in Appendix F), I think there's a good chance the others would perform differently here as well.
>
> The reviewer’s concern is fair. We are running experiments for two additional models (Pythia and Mistral) and expect to include results in the final version before the end of the discussion period.
>
> > The comparison to Shield Gemma and Llama Guard aren't exactly fair, since those are zero-shot methods. I'd like to see more baselines tuned on this particular distribution (e.g. finetuned BERT)
>
> Our choice of comparison setup is motivated by the fact that our method operates solely on the input representation and does not rely on evaluating the model’s output. In contrast, Shield Gemma and Llama Guard can incorporate iterative interactions with the model, but we restrict them to the same non-iterative, pre-output setting to ensure a fair comparison. Since our goal is specifically to evaluate safeguards that act before generation, we focus on baselines that do not depend on output-based signals. For this reason, we did not include methods that require observing or finetuning on generated outputs, as they correspond to a different class of safeguards than the one we study here. Nevertheless, we are open to try a new experiment if time allows in case we misunderstood what the reviewer was asking.
>
> >  The fact that this method is only evaluated on prompts of a specific length is a significant omission; the authors need to include results for prompts of different lengths and results averaged across all lengths.
>
> We agree with the reviewer. We have included the results of the experiments and an explanation in the general response to all reviewers, given that this was raised by more than one.
>
> > You can't claim that ID has value as a "pre-output signal" without running pre-output experiments. How do the results change when you only run the first k layers of the model under evaluation?
>
> Given that this was asked by multiple reviewers, we refer to the general response for results as a function of layer representations.
>
> Questions:
>
> > Why is Tuned Lens so much worse on jailbreak prompts compared to "malicious" and "attack" prompts? This is not mentioned in the text, but only in Table 1. Is the result a typo?
>
> We have checked again the result, and it is correct. Given the results on shorter prompts (see general response), we hypothesize this drop in performance might be due to the uneven distribution of prompt lengths in the wide 500-1000 range. In fact, for this reason, and since results for shorter prompts are stable, we are running experiments with narrower ranges also in the 500-1000 range. Results will be reported as soon as ready.
>
> > How exactly is the Tuned Lens baseline computed? Are you using the full vector of entropies at each layer as inputs to the classifier?
>
> The input to the classifier is an L-dimensional feature vector (L = number of layers), where each number corresponds to the entropy of the prompt X at layer $\ell$ ($H_{\ell}(X)$). This quantity is obtained by taking the average of the next token prediction across all the tokens in a prompt and is formally defined in Equation 14.
>
> > Do you observe the "peak" from Figure 1 on malicious prompts, or just shuffled ones?
>
> We do not observe a pronounced peak in malicious prompts. Indeed, it would have been interesting to see that malicious prompts show similarities to semantically disrupted prompts, but that is not the case. It should also be noted that these datasets are very focused on the attack task rather than being broad range prompts as the ones considered for Figure 1.

---

> > ### Comment · Reviewer_8ot4 · 2025-11-25
> >
> > Thanks for running additional experiments. Let me know when the Pythia/Mistral ones are done.

---

> > > ### Author Response · Authors · 2025-11-25
> > > **Safety on more models**
> > >
> > > We have finished running the batch of experiments, results can be found here: https://anonymous.4open.science/r/token_geometry-9DBC/reviews/rebuttal.md and they are compatible with what found for Llama 3.

---

### Official Review · Reviewer_tW55 · 2025-10-22

**Soundness:** 3
**Presentation:** 3
**Contribution:** 3
**Rating:** 6
**Confidence:** 4

**Summary:**

This paper studies the geometric properties of internal representations in  LLMs by analyzing the intrinsic dimension (ID) of the entire set of tokens within a prompt. The authors treat the evolving token representations as a mean-field particle system and measure the ID of this "token cloud" at each layer. They have three main findings: the prompt-level ID is not static, but it typically peaks in the early to middle layers of the model. Disrupting the prompt's structure by shuffling tokens causes the ID to increase, suggesting that coherent language lies on a lower-dimensional manifold. And, prompt level ID is strongly correlated with the model's uncertainty, as measured by average surprisal (next-token cross-entropy). The authors provide a theoretical intuition for this link via the geometry of the logits and the properties of the softmax function. As a practical demonstration, the authors train a simple linear probe on the vector of layer-wise ID values. This probe achieves 90-95% accuracy in distinguishing malicious from benign prompts before any text is generated, significantly outperforming existing guardrail models like Llama Guard and Gemma Shield on the tested datasets.

**Strengths:**

- I appreciate the paper's focus on the prompt-level geometry (using all tokens) rather than more common dataset-level or last-token analyses.
- The proposed feature vector (the ID at each layer) is very "low-dimensional" compared to other potential representations (e.g., all token embeddings).
- The authors don't just compare their ID probe against external safety models (Llama Guard, Gemma Shield). They also create a strong internal baseline using Tuned Lens entropy, another method for extracting layer-wise information. The fact that their geometric probe is competitive and complementary to an uncertainty-based (entropy) probe reinforces their claim that geometry is a distinct and valuable signal.

**Weaknesses:**

- The main weakness is the method's reliance on long prompts. The authors say that reliable ID estimation requires sufficiently long prompts (e.g., $N \gtrsim 500$ tokens), and their own safety experiment is filtered for prompts in the [500, 1000] token range. This limits the practical utility of the safety probe, as many (if not most) real-world malicious prompts and jailbreaks are short. The paper does not provide data on how performance degrades for shorter, more common prompts. Plus, the safety datasets were heavily filtered to a specific token range ([500, 1000]). This is highly unrepresentative of a real-world use case, which would include a vast range of prompt lengths. This filtering introduces a potential confounding variable: the probe might not just be learning "malicious vs benign" but also "properties of 500-1000 token prompts", so this makes the high accuracy less generalizable.
- The safety task is a binary classification of "malicious" vs "benign” and this is a simplification: malicious inputs can range from complex jailbreaks and prompt injections to simple toxicity or bias-inducing phrases. It is unclear if the probe is detecting a general property of "un-natural" or "adversarial" text, or if it is overfitted to the specific styles of jailbreak present in their three test datasets.
- The paper claims the ID profile is an interpretable signal, but this is only partially true. While we know the vector is separable, the paper does not show how the ID profile of a malicious prompt is different from a benign one. For instance, does the ID peak shift, increase, or decrease? The analysis showing that shuffling increases ID is clear, but the corresponding analysis for maliciousness is missing, which would be key to mechanistic insights.
- The main safety experiment (table 1), which shows the 90-95% accuracy, is only performed on the Llama 3 8B model.

**Questions:**

- Have you considered using a more complex, non-linear classifier (like a small MLP or an SVM) to see if it could capture more nuanced patterns in the ID profile?
- Have you investigated the performance of the ID-based safety probe as a function of prompt length? Specifically, how rapidly does the 90-95% accuracy degrade on shorter prompts (like $N < 100$ tokens), and have you considered modifications (like using different geometric estimators suited for sparse data or combining the ID signal with other internal metrics) to maintain high performance on these more challenging and common short-prompt cases?

---

> ### Author Response · Authors · 2025-11-21
> **Reply to referee tW55**
>
> We thank the reviewer for their constructive feedback and for recognizing the strenghts of our work. We address the point raised below.
>
> Weaknesses
>
> > The main weakness is the method's reliance on long prompts. The authors say that reliable ID estimation requires sufficiently long prompts (e.g.,  tokens), and their own safety experiment is filtered for prompts in the [500, 1000] token range. This limits the practical utility of the safety probe, as many (if not most) real-world malicious prompts and jailbreaks are short. The paper does not provide data on how performance degrades for shorter, more common prompts. Plus, the safety datasets were heavily filtered to a specific token range ([500, 1000]). This is highly unrepresentative of a real-world use case, which would include a vast range of prompt lengths. This filtering introduces a potential confounding variable: the probe might not just be learning "malicious vs benign" but also "properties of 500-1000 token prompts", so this makes the high accuracy less generalizable
>
> We include a careful reply about this point in the general response, as this was raised by multiple reviewers and it is indeed an important concern that needed to be addressed.
>
> > The safety task is a binary classification of "malicious" vs "benign” and this is a simplification: malicious inputs can range from complex jailbreaks and prompt injections to simple toxicity or bias-inducing phrases. It is unclear if the probe is detecting a general property of "un-natural" or "adversarial" text, or if it is overfitted to the specific styles of jailbreak present in their three test datasets.
>
> The reviewer raises an important point. As noted, “malicious” prompts span a wide spectrum—from jailbreaks and prompt injections to toxicity and bias-triggering phrases—which generalizes a pre‑output screening setting inherently challenging. To assess whether our probe learns generalizable properties rather than overfitting to specific jailbreak styles, we conducted the following experiments:
>
> - cross‑dataset experiments in which we train the classifier on one dataset and evaluate it on another. In this setup, performance is indeed low and often close to random.
> - cross-dataset experiments in which instead of considering the last layer ID we take up to mid-layer ID (15th layer). This is based on both our own layer‑wise analyses and prior findings in the literature showing that abstract semantic features emerge in mid‑to‑late transformer layers. This led to noticeably improved (though still quite modest) performance. See results here: https://anonymous.4open.science/r/token_geometry-9DBC/reviews/rebuttal.md
> - Train-test on the three combined datasets. Results are around 80% accuracy.
>
> > The paper claims the ID profile is an interpretable signal, but this is only partially true. While we know the vector is separable, the paper does not show how the ID profile of a malicious prompt is different from a benign one. For instance, does the ID peak shift, increase, or decrease? The analysis showing that shuffling increases ID is clear, but the corresponding analysis for maliciousness is missing, which would be key to mechanistic insights.
>
> We thank the reviewer for the insightful feedback. We are working on this and will be sending a reply soon.
>
> > The main safety experiment (table 1), which shows the 90-95% accuracy, is only performed on the Llama 3 8B model.
>
> We agree with the reviewer that only one model is too limited. Given that this point was raised by multiple reviewers, we will include new experiments in the general response as soon as they are ready.
>
> Questions
>
> > Have you considered using a more complex, non-linear classifier (like a small MLP or an SVM) to see if it could capture more nuanced patterns in the ID profile?
>
>
> We have indeed tried to use an MLP for the classifications, but didn’t see any specific raise of performance, even in the experiments where we did training on one dataset and test on another. For this reason, we keep the linear classifier results as they are more interpretable.
>
> > Have you investigated the performance of the ID-based safety probe as a function of prompt length? Specifically, how rapidly does the 90-95% accuracy degrade on shorter prompts (like  tokens), and have you considered modifications (like using different geometric estimators suited for sparse data or combining the ID signal with other internal metrics) to maintain high performance on these more challenging and common short-prompt cases?
>
> Given that more than one reviewer raised this point, we have included a set of experiments in the general response. To the best of our knowledge, the ID estimators we considered are already the state-of-the-art for sparse data. Results for shorter prompts show that the degradation with shorter prompts is not severe in most cases for ID, and moreover, the TunedLens Entropy metric performs well at any prompt length.

---

### Official Review · Reviewer_SNCD · 2025-10-26

**Soundness:** 2
**Presentation:** 1
**Contribution:** 2
**Rating:** 4
**Confidence:** 2

**Summary:**

The paper proposes a prompt-level geometry-based method to interpret the layer-wise token representation transformations through an LLM. These interpretations use the intrinsic dimension of the data manifold which is theoretically and empirically shown to be an effective representation for interpreting the workings of LLMs. The effectivess of the representation is shown with probes trained on them to classify prompts as safe/not.

**Strengths:**

1. I appreciate the perspective and utility of geometry-based methods advocated in this paper which can be used for effective prompt classification.
2. I like the constant comparison with prior related works, providing justification for the method.
3. The work is backed by theoretical soundness justifications.

**Weaknesses:**

1. My major criticism is for the presentation of the work. I think with reordering information and adding more clarifying text, this paper can be made more accessible. Specifically, I have the following recommendations. My score leans towards rejection mainly as I think this paper needs a major writing revision.
    1. Before telling the findings, the intro should describe the methods more for readers new to the mean-field theory of tokens. Example - "the intrinsic dimension exhibits a characteristic peak in early-to-middle layers." - it is not clear what the peak means here and if that's a good thing.
    2. "Empirical measure", "intrinsic dimension" should be defined early on in the intro before using these terms, but they are defined in related work.
    3. Sufficient explanation of the primary methods like GRIDE should be added in the main paper.
2. I think that just one case study for prompt safety classification, though very interesting and insightful, may not be sufficient to emphasize the utility of geometry-based methods. I suggest the following:
    1. Add another case study on a distinct interpretability task to show generality.
    2. Include a comparison with a simple probe over hidden intermediate representations entirely, to strengthen the case study for prompt safety classification.
    3. Include diverse kinds of attack prompts, e.g., prefilling attack [1,2]

## References:
1. Bypassing the Safety Training of Open-Source LLMs with Priming Attacks
2. Jailbreaking Leading Safety-Aligned LLMs with Simple Adaptive Attacks

**Questions:**

1. L157: " averaging local estimates across tokens", what's the justification for averaging as opposed to another method of aggregation?
2. L175: Why is the entropy of latent predictions for GPT-2 obtained?
8. L203: On what basis is prompt 3218 selected and how are the intermediate levels of perturbation created, eg, level 1?

---

> ### Author Response · Authors · 2025-11-21
> **Reply to reviewer SNCD**
>
> We thank the reviewer for suggesting improvements to our manuscript in a constructive way.
>
> Weaknesses
>
> > My major criticism is for the presentation of the work. I think with reordering information and adding more clarifying text, this paper can be made more accessible. Specifically, I have the following recommendations. My score leans towards rejection mainly as I think this paper needs a major writing revision.
>
> We agree with the reviewer that writing can be improved. We are currently working on modifying the text according to their suggestions. We will submit a new manuscript as soon as ready. [updated: revision now available]
>
> > I think that just one case study for prompt safety classification, though very interesting and insightful, may not be sufficient to emphasize the utility of geometry-based methods. I suggest the following:
> Add another case study on a distinct interpretability task to show generality.
>
> We thank the reviewer for the suggestion, however we doubt we will be able to include a new interpretable task and relative experiments during the rebuttal phase, both due to time, but also length, constraints.
>
> > Include a comparison with a simple probe over hidden intermediate representations entirely, to strengthen the case study for prompt safety classification.
>
> We have included an analysis of hidden intermediate layers in the general response, as it was required by multiple reviewers.
>
> > Include diverse kinds of attack prompts, e.g., prefilling attack [1,2]
>
> We thank the reviewer for suggesting a new type of attack prompts, which were not known to us. However, the cited datasets do not have sufficient data for a reliable experiment.
>
> Questions:
>
> > “averaging local estimates across tokens", what's the justification for averaging as opposed to another method of aggregation?
>
> We thank the reviewer for pointing this out. The statement “averaging local estimates across tokens” applies only to the local, pointwise estimators TLE and ESS, which return an intrinsic-dimension estimate per token and therefore require an aggregation step to obtain a prompt-level quantity. In contrast, GRIDE is a likelihood-based estimator: it directly fits a single dimension parameter by maximizing a global likelihood over the set of kNN distance ratios across all tokens, so there is no intermediate per-token ID that we average. We have updated the text in Section 3 / Appendix A to make this distinction explicit and to avoid suggesting that GRIDE uses tokenwise averaging.
> Averaging local intrinsic dimension (ID) estimates is a standard aggregation strategy in intrinsic dimension estimation. Carter et al. [1] motivate this: local homogeneity means neighborhoods estimate a shared dimension, and the arithmetic mean yields a stable global summary. For example, Cheng et al. [2] worked on last token representations, reporting the global dimension by averaging local ID estimates for both TLE and ESS. While alternatives like the harmonic mean exist for maximum-likelihood ID estimation [3], our empirical findings show that the choice between the arithmetic and harmonic mean does not significantly alter our qualitative conclusions.
>
> [1] K. M. Carter, R. Raich and A. O. Hero III, "On Local Intrinsic Dimension Estimation and Its Applications," in IEEE Transactions on Signal Processing, vol. 58, no. 2, pp. 650-663, Feb. 2010, doi: 10.1109/TSP.2009.2031722.
>
> [2] E. Cheng, C. Kervadec and M. Baroni, "Bridging Information-Theoretic and Geometric Compression in Language Models," in Proceedings of the 2023 Conference on Empirical Methods in Natural Language Processing (EMNLP), Singapore, Dec. 2023, pp. 12397-12420, doi: 10.18653/v1/2023.emnlp-main.762.
>
> [3] D. MacKay. and Z. Ghahramani. Comments on ‘Maximum Likelihood Estimation of Intrinsic Dimension’ by E. Levina and P. Bickel (2004). Comment on personal webpage (2005)
>
> >  Why is the entropy of latent predictions for GPT-2 obtained?
>
> We compute the entropy of the latent predictions across several major model suites: GPT-2, Pythia, Llama, and OPT. These models form a comprehensive set of those for which trained linear probes are already available at https://huggingface.co/spaces/AlignmentResearch/tuned-lens/tree/main/lens.
>
> > On what basis is prompt 3218 selected and how are the intermediate levels of perturbation created, eg, level 1?
>
> The prompt is selected randomly. We observe similar features on different prompts. The general behaviour is shown through the average over prompts. An explanation of the intermediate levels of perturbation is present at appendix B (see e.g. fig. 7)

---

> > ### Author Response · Authors · 2025-11-24
> > **Updated Manuscript with text revisions as suggested by SNCD**
> >
> > We have now updated the manuscript with a rephrasing of the introduction and methodology sections to
> > - introducing concepts intuitively in the introduction and extensively in the methodology section, to allow the reader to better understand the motivation of the paper
> > - included definitions of ID estimators in methodology section

---

### Official Review · Reviewer_NKRy · 2025-10-31

**Soundness:** 2
**Presentation:** 1
**Contribution:** 1
**Rating:** 4
**Confidence:** 2

**Summary:**

The paper introduces a geometric lens, a prompt-level intrinsic dimension, to probe internal LLM dynamics, offering both mechanistic insights and a practical safety tool.  Treating transformers as mean-field particle systems, the authors estimate ID via k-nearest-neighbor estimators (GRIDE, ESS, TLE) for every layer and every prompt.
The authors demonstrate that this prompt-level ID, estimated layer by layer, exhibits a strong positive correlation with the model's next-token uncertainty (average surprisal).  Key findings indicate that the ID peak occurs in early-to-middle layers and increases when the prompt's semantic structure is disrupted by shuffling.  As a practical case study, the authors use the per-layer ID profile as a feature vector to train a linear probe for detecting malicious prompts before generation.
This method achieves 90-95% accuracy, reportedly outperforming standard guardrail models like Llama Guard and Shield Gemma on the tested datasets.

**Strengths:**

1. This paper introduces a novel methodology, using a geometric probe for internal representations.
- This paper introduces a prompt-level intrinsic dimension rather than a last-token or dataset-level analysis.
- This paper clarifies the distinction between prompt-level and dataset-level ID, resolving apparent contradictions in prior work.
- This paper provides a bridge between geometric structure and information-theoretic uncertainty.

2. This paper introduces applications of practical interpretability and safety applications.
- This paper demonstrates that a simple linear probe on ID profiles achieves 90–95% accuracy in malicious prompt detection. It outperforms established guardrails (Llama Guard, ShieldGemma) on the same data splits
- The experiment shows complementarity with TunedLens entropy features, suggesting geometric and uncertainty signals are aligned.

3. This paper has rigorous experimental validation across models and estimators. It uses four major LLMs (LLaMA, Mistral, Pythia, OPT) and three distinct ID estimators (GRIDE, ESS, TLE).

**Weaknesses:**

1. There are some contradictory findings in latent entropy analysis. The paper's central thesis is that geometry (ID) tracks uncertainty (entropy). However, the analysis in Appendix F (correlating ID with latent entropy from TunedLens) contradicts this for some of the main models studied. While GPT-2 and Pythia models show the expected positive correlation, the Llama 3 and OPT models show a negative correlation in the middle-to-late layers.

2. The experiments are based on kNN assumptions and hyperparameters.  The kNN-based ID estimators assume local uniform density, which may not hold in high-curvature manifolds. The ID values vary with range scaling and neighborhood size, implying estimator sensitivity.

3. The experiments have ambiguity in the shuffling methodology and its semantic impact.  There is no quantitative measure of semantic disruption (e.g., BLEU, BERTScore) provided to calibrate shuffle severity.   The claim that shuffling “disrupts syntax and semantics” is asserted but not measured; only unigram frequency is preserved.

4. The experiments have a limitation on the prompt length and domain constraints.  The prompts used in this paper, the length $N$ is not less than 1024 in Section 3.1. There is limited usability for short prompts, conversational inputs, or non-text modalities.  This reduces the method’s coverage in realistic LLM interactions.

**Questions:**

-  How sensitive are ID estimates to prompt length, especially near the 500-token lower bound used in safety experiments (Section 6)?
-  How does block-wise shuffling compare to full random permutation in terms of ID increase?
-  Why use shuffling? Since shuffling would change the semantic information of the prompt, making it out of distribution.
- The empirical correlation between Logit ID and Contextual Entropy is $\rho=0.60$ (Fig. 3). What other geometric or non-geometric factors are hypothesized to account for the remaining variance?
- What is the computational cost of per-layer ID extraction for large-scale deployment?

---

> ### Author Response · Authors · 2025-11-21
> **Reply to reviewer NKRy**
>
> We thank the reviewer for their constructive feedback. We address the point raised below. Some replies are included in the general response as they address multiple reviewers.
>
> Weaknesses
>
> > There are some contradictory findings in latent entropy analysis. The paper's central thesis is that geometry (ID) tracks uncertainty (entropy). However, the analysis in Appendix F (correlating ID with latent entropy from TunedLens) contradicts this for some of the main models studied. While GPT-2 and Pythia models show the expected positive correlation, the Llama 3 and OPT models show a negative correlation in the middle-to-late layers.
>
> We thank the reviewer for raising this point. One of the main findings of the paper is indeed the correlation between ID and entropy, as shown in Figure 2 in Section 5, which holds across models, including Llama. Since this is an empirical finding, we attempt to explain it by making some assumptions, as described in the paragraph “Why does ID track uncertainty.” We illustrate this through the following sequence of operations:
> $$
> \mathrm{ID}(\text{last layer embedding} \ X) \xrightarrow{\text{unembedding}} \mathrm{ID}(\text{logits}) \xrightarrow{\text{softmax}} \mathcal{H}(X)
> $$
> where $\mathcal{H}(X)$ is the average contextual entropy of the model prediction.
>
> While we consistently observe a high correlation (without exceptions) between the last-layer embedding and the softmax entropy of the final next-token prediction, this relationship does not always hold when unembedding representations from intermediate layers. In particular, we find that the negative correlation arises at the step:
> $$
> \mathrm{ID}(\text{intermediate logits})
> \xrightarrow{\text{softmax}}
> \mathcal{H}(X_\ell))
> $$
> where $\mathcal{H}(X_\ell)$ is the average contextual entropy of the intermediate prediction at layer $\ell$.
>
> We believe this occurs because certain conditions required for the positive correlation do not hold in these cases. Further analysis is necessary to identify the assumptions under which we can expect a positive correlation. We speculate that the relationship emerges only once representations are sufficiently learned for the correlation to hold, since we also do not observe a positive correlation in untrained models.
>
> > The experiments are based on kNN assumptions and hyperparameters. The kNN-based ID estimators assume local uniform density, which may not hold in high-curvature manifolds. The ID values vary with range scaling and neighborhood size, implying estimator sensitivity.
>
> We provide an analysis of the assumptions of our estimators in Appendix A.
>
> > The experiments have ambiguity in the shuffling methodology and its semantic impact. There is no quantitative measure of semantic disruption (e.g., BLEU, BERTScore) provided to calibrate shuffle severity. The claim that shuffling “disrupts syntax and semantics” is asserted but not measured; only unigram frequency is preserved.
>
> In this paper, we quantify the degree of shuffling using the shuffle index described in the ‘Shuffling method’ paragraph. We find that the shuffle index is correlated to quantitative measures of semantic disruption (BLEU, BERTScore) as shown in the table in https://anonymous.4open.science/r/token_geometry-9DBC/reviews/rebuttal.md. The analysis uses 50 prompts from Pile-10K with the Llama-3-8B tokenizer. The results show a consistent drop in both BLEU and BERTScore across shuffle degrees. BLEU drops from 1.00 to 0.03, while BERTScore decreases from 1.00 to 0.76, quantifying the semantic disruption caused by token shuffling at each degree.
>
> > The experiments have a limitation on the prompt length and domain constraints. The prompts used in this paper, the length  is not less than 1024 in Section 3.1. There is limited usability for short prompts, conversational inputs, or non-text modalities. This reduces the method’s coverage in realistic LLM interactions.
>
> Given that this point was raised by multiple reviewers, we have run a set of experiments and provided results in the general response.

---

> > ### Author Response · Authors · 2025-11-21
> > **part two**
> >
> > Questions:
> >
> > > How sensitive are ID estimates to prompt length, especially near the 500-token lower bound used in safety experiments (Section 6)?
> >
> > Given that this question was raised by more than one reviewer, we addressed it in the general response.
> >
> > >  How does block-wise shuffling compare to full random permutation in terms of ID increase?
> >
> > We do perform full random permutation in our highest level of shuffling (S=5). We refer to Appendix B (e.g. fig 7) for an explanation of the shuffling algorithm.
> >
> > > Why use shuffling? Since shuffling would change the semantic information of the prompt, making it out of distribution.
> >
> > The goal of shuffling is to show that ID at token level is sensitive to the semantic disruption of the input, and it therefore carries relevant information on the model's internal workings.
> >
> > > The empirical correlation between Logit ID and Contextual Entropy is  (Fig. 3). What other geometric or non-geometric factors are hypothesized to account for the remaining variance?
> >
> > Figure 3 captures a leading-order empirical trend that we can motivate in a toy setting: if the next-token distribution has a uniform effective support over a $D_{\mathcal M}$-dimensional logit manifold, then the expected contextual entropy scales as $\langle S\rangle \sim \log D_{\mathcal M}$ as shown in Equation 9. The residual variance is expected in real models, where these assumptions fail; the support for next token prediction is typically anisotropic and non-uniform (e.g., clustered or sparse over a subset of directions), and the logit manifold can be curved, all of which alter entropy at fixed ID.

---

> > ### Comment · Reviewer_NKRy · 2025-11-24
> > **Further concern about the Weakness**
> >
> > Thanks for providing the experiments about the semantic disruption.
> >
> > As the results show, shuffle will change the semantic information, which is expected.  In Figure 7, if a sentence is a totally random sequence, it could be assumed that this input is random noise or out-of-distribution. In this case, it is unclear whether the uncertainty measurement is reasonable. In general, the random noise should have a high $H$.  One goal of this paper is to discuss the relationship between ID and Uncertainty. I believe the experimental setting is not fair enough. It is better to use some in-distributed prompts with different IDs and the same semantic information.  Then the correlation between ID and Uncertainty makes sense.
> >
> > In Figure 2, the correlation does not hold as the layer goes deeper. The paper does not provide the entropy and IDs for each layer.   Does the entropy in each layer contain the same? Do IDs vary a lot across different layers?
> >
> > In Section 5,  the author mentioned "**Why does intrinsic dimension track uncertainty?** The observed correlation between a geometric quantity (the ID of internal representations) and an information-theoretic quantity (the surprisal) occurs at the softmax layer between the last layer representations and the next token predictions", however, in Figure 2, the deeper layer have a lower correlation, which is not consistent with this part. In Line 340, I believe some ID should be replaced by Tokens.

---

> > > ### Author Response · Authors · 2025-11-25
> > > **Reply to further concerns about weakness**
> > >
> > > We thank reviewer NKRy for taking the time to reply to our comments.
> > >
> > > > As the results show, shuffle will change the semantic information, which is expected. In Figure 7, if a sentence is a totally random sequence, it could be assumed that this input is random noise or out-of-distribution. In this case, it is unclear whether the uncertainty measurement is reasonable. In general, the random noise should have a high $H$. One goal of this paper is to discuss the relationship between ID and Uncertainty. I believe the experimental setting is not fair enough. It is better to use some in-distributed prompts with different IDs and the same semantic information. Then the correlation between ID and Uncertainty makes sense.
> > >
> > > We believe the reviewer might have partially misunderstood our claims, so we clarify here, and we are open to modifying the draft accordingly if needed. The measure of correlation between ID and surprisal is done *in-distribution*, i.e. on tokens which have not been shuffled, as the shuffling is discussed in the previous section.
> > >
> > > Section 4, which discusses the shuffling, has the goal of qualitatively describing the ID of prompts at token level. The use of shuffling for this qualitative description is not new in the literature and indeed it was used in previous work where ID was calculated at the prompt level (on last layer representations). Indeed a secondary goal of this section is to compare the different approaches of this work (token level) and previous work (prompt level). We dedicate an in-depth comparison in Appendix D.
> > >
> > > Section 5, on the other hand, shows the empirical finding that ID is correlated to surprisal, and this calculation is not done by shuffling sentences.
> > >
> > > > In Figure 2, the correlation does not hold as the layer goes deeper.
> > >
> > > A correlation of rho>0.6 is considered significant for the measure we use (Pearson correlation), thus we believe that the correlation also holds at the last layers.
> > >
> > > > The paper does not provide the entropy and IDs for each layer. Does the entropy in each layer contain the same? Do IDs vary a lot across different layers?
> > >
> > > As per eq. 3, average surprisal is computed once per model per prompt, not at every layer. ID on the other hand can be computed at each layer. Hence, fig. 2 shows correlation as a function of layer where correlation has been computed pairing ID at a given layer and average surprisal for that prompt.
> > >
> > > > In Section 5, the author mentioned "Why does intrinsic dimension track uncertainty? The observed correlation between a geometric quantity (the ID of internal representations) and an information-theoretic quantity (the surprisal) occurs at the softmax layer between the last layer representations and the next token predictions", however, in Figure 2, the deeper layer have a lower correlation, which is not consistent with this part.
> > >
> > > A correlation rho>0.6 is considered significant for Pearson correlation, so the statement holds. The fact that correlation decreases as a function of layer is not addressed in our explanation.
> > >
> > > > In Line 340, I believe some ID should be replaced by Tokens.
> > >
> > > We propose to change the sentence to: "We expect the ID of the last layer embedding to be strongly correlated[...]" for better clarity.

---

> > > > ### Comment · Reviewer_NKRy · 2025-11-27
> > > > **Reply to authors**
> > > >
> > > > Thank you for the additional clarification. To confirm, Figure 2 is based on the original prompts, whereas Figure 1 focuses on the shuffling method.  However, the roles of Section 4 and Appendix C remain unclear. Their contributions to the paper are difficult to recognize, and it would be helpful for the authors to clarify how these sections support the core empirical findings.

---

> ### Author Response · Authors · 2025-11-27
> **Reply to reviewer NKRy**
>
> We thank NKRy for keeping the discussion active, we appreciate this.
>
> > To confirm, Figure 2 is based on the original prompts, whereas Figure 1 focuses on the shuffling method.
>
> Yes.
>
> > However, the roles of Section 4 and Appendix C remain unclear. Their contributions to the paper are difficult to recognize, and it would be helpful for the authors to clarify how these sections support the core empirical findings.
>
> Section 4 and Appendix C play a specific role in the paper, distinct from the core empirical finding of Section 5. Their purpose is to give a qualitative, model‑agnostic picture of prompt‑level geometry itself, before introducing its connection to surprisal. Section 4 shows that intrinsic dimension exhibits a stable and interpretable layerwise profile and that this profile shifts predictably under controlled perturbations (shuffle), establishing that prompt‑level ID is a meaningful geometric observable. Appendix C complements this by showing that the qualitative effect of shuffling is consistent across models, prompts, and ID estimators. These sections therefore provide the empirical motivation and basic phenomenology that Section 5 builds on: Section 5 does not depend on shuffling, but uses unshuffled prompts only; however, the reader first needs to see that prompt‑level ID is a stable, well‑behaved quantity whose differences are not estimator‑ or model‑artifacts.
> Another point that is significant for us is the comparison to previous literature, to highlight the difference with the prompt-level analyses, which is what previous work has focused on when looking at the geometry of internal representations with a similar approach. Appendix D expands on this.
>
> We propose a modification to the draft to guide the reader through our thought process better:
>
> [addition to the first paragraph of section 4, after "[...] across layers for different models."]
>
> “The goal of this section is to establish that prompt‑level intrinsic dimension is a stable geometric observable and that its layerwise evolution exhibits consistent, interpretable structure across models. The shuffling experiment is used as a controlled perturbation to illustrate how prompt geometry responds to syntactic and semantic disruption. Appendix C extends the qualitative checks of this section and confirms their robustness across models and estimators.”
>
> Additionally, we add a footnote at the beginning of section 5, after "[...]surprisal of the next token prediction":
>
> "Differently from the previous section, ID in this section is computed exclusively on unshuffled prompts."

---

### Author Response · Authors · 2025-11-21
**General reply to questions shared across different reviewers**

We thank all the reviewers for their constructive and useful feedback. We have synthesized a few suggestions that recurred across multiple reviewers. These are points that strengthen the paper, so we are dedicating time to run the necessary experiments. To keep the discussion moving, we report a first round of results and outline the remaining experiments planned for early next week.

The reviewers converge on several strengths of the manuscript:

- using prompt-level intrinsic dimension as a geometric lens on internal LLM behavior.
- Strong empirical grounding, with thorough comparisons across models and ID estimators.
- A clear link between geometric structure and uncertainty,
- a demonstration that ID-based features are competitive with, and complementary to, TunedLens entropy features.

Conversely, the reviewers agree that the evaluation of the safety task could be improved. They suggested to

- include shorter prompts to assess the scalability of ID estimates;
- test internal representations across multiple layers, not just the last layer;
- consider more than one language model.

Here we report our experiments to answer to these three points.

1. Shorter prompts

In the first submission, the reason for running long prompts was to ensure stability of the ID estimators. Nevertheless, we took the reviewer's suggestion to evaluate on shorter prompts, also as a way to empirically check the actual limit at which the reliability of the method degrades.

We run the pipeline for ranges of 100-200, 200-300, 300-400 and 400-500 prompt length, both for the GRIDE ID estimator and for the TunedLens Entropy estimator. All results are detailed here: https://anonymous.4open.science/r/token_geometry-9DBC/reviews/rebuttal.md

In summary, we find that:

- degradation in the ID estimator is low for most cases even for short prompts
- the entropy estimator does not degrade in performance at all, given it does not need large point clouds for a good estimate
- Narrower ranges seem to be more stable, as the distribution of prompt length has less variance.

Given these results, we have run experiments for 500-600 and are open to substitute these results to the wider 500-1000 range originally included in the submission, as their are more reliable, and cover a more realistic range of prompt lengths.

2. Internal Representations

Some reviewers suggest it might be interesting to check whether internal layers might also be able to discriminate malicious from benevolent prompts. This hypothesis is also supported by various existing results in LLM interpretability, claiming that middle layers already possess a degree of understanding of abstract semantic information in the prompt. We quote results here: https://anonymous.4open.science/r/token_geometry-9DBC/reviews/rebuttal.md
As we can see, performance rises at around mid-layers both for GRIDE and Entropy metrics, confirming our expectations.

3. Safety task on more models

We are have run the safety pipeline to Pythia and Mistral, given they have been used for ID analyses in the paper. [results now available here: https://anonymous.4open.science/r/token_geometry-9DBC/reviews/rebuttal.md]

---

### Author Response · Authors · 2025-11-25
**Safety task on more models**

We have run our malicious-benign pre-output detection pipeline on Mistral 7B and Pythia 6.9B for one of the datasets with prompt length 300-400 (so shorter prompts than what quoted in the original submission). Results can be found in section "Results on Pythia 6.9B and Mistral 7B" in Results on Pythia 6.9B and Mistral 7B. Results are compatible with what was found for Llama 3, see section "Shorter Prompts" in https://anonymous.4open.science/r/token_geometry-9DBC/reviews/rebuttal.md.

---

### Author Response · Authors · 2025-12-03
**Summary of Rebuttal Status & Guide to Revisions**

To the newly assigned Area Chair,

We understand your challenge of reviewing many papers in a short amount of time. For that reason, we provide a summary of the rebuttal so far.

First of all, we would like to thank all reviewers for providing constructive and reasonable feedback; it has significantly helped improve our work.

We now summarise the comments raised by the reviewers. There were basic requests for missing evaluations that needed to be addressed promptly, namely:

- Efficiency of ID estimators
- Wider range of prompt lengths in the downstream task
- Use of internal representations (not just the last layer) as probes for the downstream task
- Use of more than one language model for the downstream task

We have successfully executed these experiments, as shown in the responses below. These experiments further support the paper's claims, and we would be happy to include them in the final version.

There were a few other reviewer-specific concerns. For brevity, we summarize them below, including our responses.

- Reviewer NKRy claimed there were contradictory findings in the latent entropy analysis. Through a constructive discussion, we have clarified that there are no contradictions and resolved the misunderstanding by modifying the text accordingly.

- Reviewer NKRy asked about details on the hyperparameters of the pipeline. We pointed at the relative appendix.

- Reviewer NKRy was concerned about quantitatively measuring semantic disruption by shuffling. We provided this quantification using the method suggested.

- Reviewer SNCD requested a revision of the introduction and method sections for clarity. We provided a rephrasing following their suggestion.

- Reviewer SNCD asked for a second downstream task. We have kindly declined the request because of the limited time in the rebuttal phase.

- Reviewer SNCD suggested another type of dataset for the downstream task. Upon inspection of the dataset, we kindly declined the request due to a lack of sufficient data in the dataset for a reliable analysis.

- Reviewer tW55 raises concerns about the generalization of our classifier. We run experiments to test this point, and we would be happy to include the results in the final version. We refer the AC to the full response for more details.

We believe we addressed all concerns that reasonably provide a cause for rejection; thus, we hope for a positive evaluation by the AC.

---

### Meta-Review · Area_Chair_sbGE · 2026-01-06

**Summary:**

This paper studies prompt-level intrinsic dimension of token representations across transformer layers, linking geometric structure in internal representations to model uncertainty, and proposing a lightweight pre-output safety signal based on layerwise ID profiles.

Reviewers generally agree that the paper introduces a novel and interesting perspective. The main concerns focus on presentation and evaluation. First, clarity and organization were recurring issues: multiple reviewers found the narrative difficult to follow, with key concepts introduced too late or without sufficient intuition. In particular, the distinction between the shuffling experiments and the ID–uncertainty correlation analysis was initially unclear. Second, reviewers raised concerns about the experimental scope and fairness of the safety evaluation, especially the reliance on long prompts and a single model in the original submission. In response, the authors added experiments covering a wider range of prompt lengths and extended the safety results to additional models. However, generalization across datasets remains limited.

**Reviewer Concerns:**

The rebuttal addressed several major reviewer concerns. Issues related to prompt length sensitivity, reliance on a single model, lack of efficiency analysis, and the use of only final-layer representations in the safety task were substantially addressed through additional experiments (shorter prompts, multiple models, layerwise analysis, and estimator efficiency). Concerns about ambiguity in the shuffling experiments and their role relative to the ID–uncertainty correlation were clarified.

Some concerns remain partially outstanding. In particular, generalization of the safety probe across datasets remains limited, and the method’s behavior under realistic deployment constraints could be framed more clearly. While the explanation of intermediate-layer entropy inconsistencies is plausible, it may not fully resolve all conceptual concerns raised.

**Reviewer Scores:**

Reviewer NKRy and 8ot4: Likely improved modestly.

---

### Decision · Program_Chairs · 2026-01-26

Reject